# Observation of interband Berry phase in laser-driven crystals

Ayelet J. Uzan-Narovlansky[1,2,10 ✉], Lior Faeyrman[1,10], Graham G. Brown[3], Sergei Shames[1], Vladimir Narovlansky[4], Jiewen Xiao[5], Talya Arusi-Parpar[1], Omer Kneller[1], Barry D. Bruner[1], Olga Smirnova[3,6], Rui E. F. Silva[7], Binghai Yan[5], Álvaro Jiménez-Galán[3,7], Misha Ivanov[3,8,9] & Nirit Dudovich[1 ✉]

Ever since its discovery[1], the notion of the Berry phase has permeated all branches of physics and plays an important part in a variety of quantum phenomena[2]. However, so far all its realizations have been based on a continuous evolution of the quantum state, following a cyclic path. Here we introduce and demonstrate a conceptually new manifestation of the Berry phase in light-driven crystals, in which the electronic wavefunction accumulates a geometric phase during a discrete evolution between different bands, while preserving the coherence of the process. We experimentally reveal this phase by using a strong laser field to engineer an internal interferometer, induced during less than one cycle of the driving field, which maps the phase onto the emission of higher-order harmonics. Our work provides an opportunity for the study of geometric phases, leading to a variety of observations in light-driven topological phenomena and attosecond solid-state physics.

Whenever a quantum system undergoes a cyclic evolution governed by a change of parameters, it acquires a phase factor, known as the geometric phase. The most common formulations of the geometric phase are the Aharonov–Bohm phase[3] and the Berry phase[1]. Over the past several decades, the geometric phase has been generalized and became notable in several applications—from condensed matter physics[4,5], fluid mechanics[6] and optics[7,8] to particle physics and gravity[9].

In condensed matter physics, the geometric phase manifests in the electronic Bloch states, leading to various observations such as the quantum Hall effect, electric polarization, orbital magnetism and exchange statistics[4]. In these systems, applying an electric field drives the electronic wavefunction in the crystal momentum space, leading to the accumulation of the Berry phase because of the parameter space topology, and it is known as Zak's phase when integrated over the entire Brillouin zone[10]. The local properties of this quantum evolution are captured by the Berry curvature, representing the local rotation of the wavepacket as it evolves within the Brillouin zone. The original description of Berry's phase[1] required two fundamental conditions. First, the phase should be accumulated as a quantum state evolves in a parameter space adiabatically. Second, the parameter should be modified continuously. A generalization of the Berry phase[11,12] removed the adiabaticity requirement. However, the smooth modification of the wavefunction in a continuous parameter space, which underlies the basic mathematical formulations of the Berry phase, forms the main part of its various realizations[13].

Here we introduce and experimentally verify a formulation of the geometric phase, which includes both continuous and discrete modifications of the wavefunction. This phase, referred to as the interband Berry phase, is pertinent to all light-driven quantum systems undergoing both adiabatic evolution and light-induced jumps in the Hilbert space. Experimentally, we focus on the light-driven condensed matter systems. Driven by a low-frequency external field, the electronic wavefunction undergoes non-adiabatic interband transitions followed by intraband propagation and, finally, an additional non-adiabatic transition by photo-recombination. These dynamics form a closed loop in the energy–momentum space (Fig. 1a). Although the evolution of the wavefunction in each band is continuous, the light-induced transitions between the bands represent a discrete evolution. The geometric phase accumulated along this closed path is gauge invariant[14] measurable and plays an important part in the response of a quantum system to an intense light field (see a detailed discussion in the Supplementary Information).

We resolve the interband Berry phase[15] by introducing attosecond interferometry, using a polarization-controlled laser field to drive the evolution of the quantum wavefunction. Our scheme induces an internal interferometer in the $k$-space by shaping the electronic trajectories on a subcycle time scale, providing access to the interband Berry phase. By manipulating the instantaneous polarization of the laser field, we induce and control two different electron–hole paths, evolving during the positive and negative subcycles of the laser field. Their phase difference, recorded in a broken-inversion-symmetry crystal as a function of laser field polarization, is resolved using high-harmonic generation (HHG) spectroscopy[16,17] (Fig. 1b). Driven by the strong laser field[18], the electron tunnels across the energy gap between the valence and the conduction bands, initiating an electron–hole wavepacket[19]. This excitation is followed by the propagation of

[1]Department of Complex Systems, Weizmann Institute of Science, Rehovot, Israel. [2]Department of Physics, Princeton University, Princeton, NJ, USA. [3]Max-Born-Institut, Berlin, Germany. [4]Princeton Center for Theoretical Science, Princeton University, Princeton, NJ, USA. [5]Department of Condensed Matter, Weizmann Institute of Science, Rehovot, Israel. [6]Technische Universität Berlin, Ernst-Ruska-Gebäude, Berlin, Germany. [7]Instituto de Ciencia de Materiales de Madrid (ICMM), Consejo Superior de Investigaciones Científicas (CSIC), Madrid, Spain. [8]Blackett Laboratory, Imperial College London, London, UK. [9]Department of Physics, Humboldt University, Berlin, Germany. [10]These authors contributed equally: Ayelet J. Uzan-Narovlansky, Lior Faeyrman. ✉e-mail: auzan@princeton.edu; nirit.dudovich@weizmann.ac.il

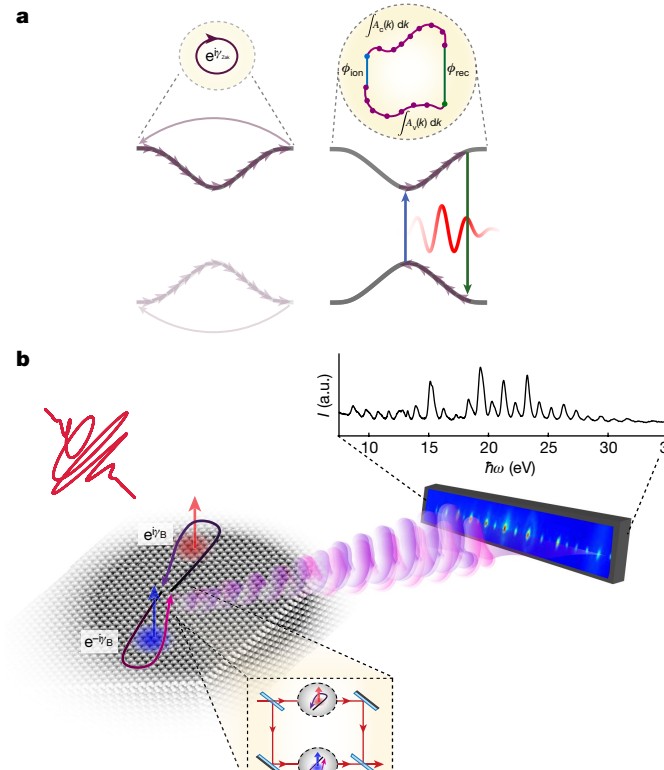

**Fig. 1 | Interband Berry phase resolved using HHG spectroscopy.**
**a**, The Berry phase in condensed matter systems. Left, intraband Berry phase, accumulated as the wavepacket, continuously evolves in the *k*-space within the band. When the trajectory forms a closed loop through the entire Brillouin zone, a gauge-invariant phase is accumulated (known as the Zak phase). Right, interband Berry phase, the wavepacket evolution includes discrete transitions between the two bands, closing a loop in the energy–momentum space. **b**, HHG Berry-phase interferometry. An interferometric measurement is performed by manipulating the instantaneous polarization of the laser field, generating two different wavepackets, evolving along the positive and the negative half cycles. The interference pattern is resolved in the HHG spectrum, encoding the relative accumulated Berry phase.

the electron–hole wavepacket, dictated by the temporal shape of the laser field, and electron–hole recombination, projecting the *k*-space trajectories onto the emission of higher-order harmonics (known as interband HHG)[20,21]. The Berry phase accumulated by the electron–hole wavepacket is thus mapped onto the optical phase and amplitude of the emitted harmonics. As various *k*-space trajectories are projected onto different harmonics[22,23], this scheme can resolve the evolution of the Berry phase over the entire Brillouin zone. Finally, we obtain a direct insight into the local manifestation of the geometrical properties of the wavefunction, the Berry curvature[24–27], resolving its impact on the electron currents.

The primary advantage of HHG spectroscopy lies in its time scale—the entire interaction evolves during less than one optical cycle, avoiding scattering or dephasing events and preserving the coherence of the wavepacket[22,28]. A previous study[25] showed the Berry curvature of topological insulators using HHG driven by THz field[29], having a fundamental period of 40 fs. In their study, topology helps to overcome dephasing and scattering mechanisms, showing the geometrical properties of the system. Our measurement, performed on an attosecond time scale, enables the probing of the Berry phase in trivial insulators.

Formally, the gauge-invariant geometric phase accumulated during a cyclic evolution of the wavefunction in the energy–momentum space can be evaluated as follows:

$$\lim_{N\to\infty}\langle u_{v,\mathbf{k}_1}|u_{c,\mathbf{k}_2}\rangle\langle u_{c,\mathbf{k}_2}|u_{c,\mathbf{k}_3}\rangle\cdots\langle u_{c,\mathbf{k}_{N-1}}|u_{v,u_N}\rangle\cdots\langle u_{v,\mathbf{k}_2}|u_{v,\mathbf{k}_1}\rangle$$

$$\propto e^{i\int_{t'}^{t}\varepsilon_g(\mathbf{k}(\tau))d\tau+i\gamma_{B,int}} \tag{1}$$

$$\gamma_{B,int}\equiv\int_{t'}^{t}\mathbf{F}(\tau)\cdot(\mathcal{A}_g(\mathbf{k}(\tau))+\nabla_\mathbf{k}\phi_d(\mathbf{k}(\tau)))d\tau$$

Here $|u_{n,\mathbf{k}}\rangle$ is the periodic part of the Bloch function ($n$ = v, c for valence and conduction bands), $\varepsilon_g=\varepsilon_c-\varepsilon_v$ is band gap energy and $\mathcal{A}_g=\mathcal{A}_c-\mathcal{A}_v$ is electron–hole relative Berry connection $\mathcal{A}_n(\mathbf{k})=(i\langle u_{n,\mathbf{k}}(x)|\nabla_\mathbf{k}|u_{n,\mathbf{k}}(x)\rangle)$. The discrete evolution between the bands is described by the phase of the interband dipole coupling, $\phi_d(\mathbf{k})=\arg(i\langle u_{v,\mathbf{k}}(x)|\nabla_\mathbf{k}|u_{c,\mathbf{k}}(x)\rangle)$. The crystal quasi-momentum, $\mathbf{k}(\tau)=\mathbf{k}-\mathbf{A}(t)+\mathbf{A}(\tau)$ is controlled by the laser field, where $\mathbf{F}(t)$ and $\mathbf{A}(t)$ are the laser electric field and the vector potential, respectively. The instants $t'$ and $t$ define the transition times between the bands (the ionization and recombination times). The interband Berry phase, $\gamma_{B,int}$, contains the evolution inside each band, described by the conventional integral over the Berry connection, together with the phase contributions associated with the jumps between the bands, which are represented by the phases of the coupling dipoles. This Berry phase represents a closed trajectory in energy–momentum space and can be expressed as $\int_{\mathbf{k}_i}^{\mathbf{k}_f}[\mathcal{A}_c(\mathbf{k})-\mathcal{A}_v(\mathbf{k})]d\mathbf{k}-(\phi_d(\mathbf{k}_f)-\phi_d(\mathbf{k}_i))$. Recombination maps each closed trajectory into the emission of optical radiation, at a frequency of $\varepsilon_g(\mathbf{k}(t))$, projecting the Berry phase, $\gamma_{B,int}$, onto the optical phase of the emitted harmonics.

**Interband Berry-phase interferometry**
We extract the Berry phase using the sub-laser-cycle interferometric measurement. The two arms of the interferometer are the two electron trajectories, which are inverted with respect to each other and evolve during the positive and negative half cycles of the laser field (Fig. 1b). By controlling the instantaneous laser-field polarization, we control these trajectories in the *k*-space and manipulate their interference. The control is achieved using elliptically polarized light, which induces the two-dimensional (2D) motion in the *k*-space. When the two trajectories evolve in the vicinity of positive and negative Berry curvatures, the accumulated phases along the two arms will have opposite signs (Fig. 2a). Increasing the ellipticity, $\epsilon$, enables us to continuously tune the 2D *k*-space paths, and therefore the accumulated Berry phase. Finally, the two emission bursts, associated with the radiative recombination of the two trajectories, interfere in the HHG spectrum, encoding their relative phase in the spectral shape of the harmonics.

We experimentally demonstrate the Berry-phase interferometry by producing HHG from an α-quartz z-cut crystal[24,30,31], using a 1.2-μm laser field with an intensity of order of $10^{13}$ W cm$^{-2}$. The harmonics spectrum spans up to 30 eV, enabling us to probe the internal dynamics over a large energy range (Fig. 1b). By performing detailed theoretical and experimental studies (Supplementary Information), we conclude that under our experimental conditions, the interband mechanism dominates the harmonics emission. We note that this observation is in contrast to the previous observation of HHG in quartz[30], performed with shorter wavelength and laser pulses of few cycles.

Figure 2b,c presents the HHG signal as a function of the driving field ellipticity ($\epsilon$) along the Γ−K and Γ−M axes. Along the Γ−K axis, the harmonic signal decreases as the ellipticity is increased (Fig. 2b). Owing to the rotation symmetry (C2) of the crystal along this axis, an electron trajectory, modified by the ellipticity of the field, does not accumulate an additional Berry phase (Supplementary Information). In this case, as we increase the ellipticity, the transverse momentum increases, leading to the suppression of the electron–hole recombination. Rotating the crystal to the Γ−M axis leads to a different response. Along this axis, increasing the ellipticity decreases the odd harmonics signal and increases the even harmonics signal (Fig. 2c). For high ellipticity values,

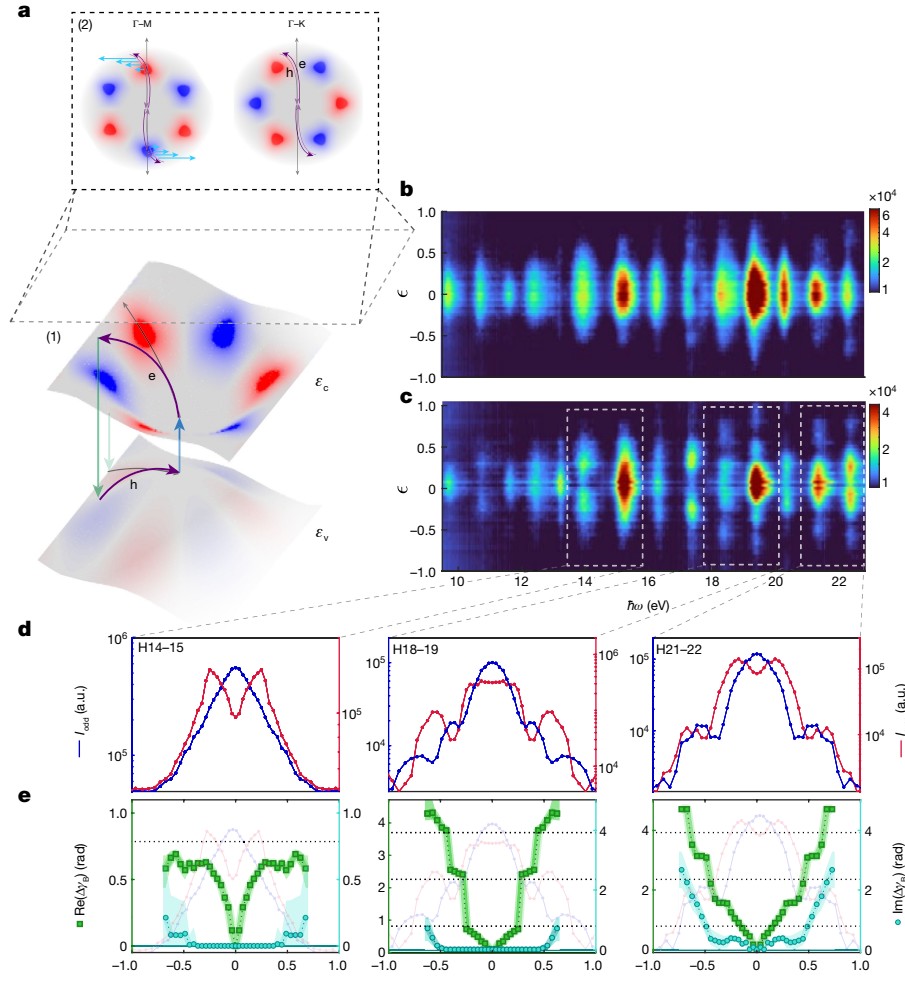

**Fig. 2 | Berry-phase interferometry. a**, Schematic of the electron trajectories driven by an elliptical field: (1) manipulating the laser ellipticity, $\epsilon$, induces a transversal evolution of the electron–hole wavepacket, controlling the closed loop in energy–momentum space. $\varepsilon_v$ and $\varepsilon_c$ correspond to the valance and conduction bands, respectively, coloured according to the Berry curvature (red representing positive values and blue representing negative values). Black arrows correspond to the trajectory induced by the linear driving field and purple arrows to the elliptical driving field; (2) a top projection of the energy–momentum space, $\varepsilon_c - \varepsilon_v$, illustrates the intraband wavepacket evolution along the Γ−*M* (30° crystal orientation) and Γ−K (0° crystal orientation). The light-blue arrows represent the Berry connection increasing along the transverse evolution, leading to the accumulation of the Berry phase. **b,c**, HHG spectrum (log scale) as a function of the driving field ellipticity, resolved along the Γ−K (**b**) and Γ−M (**c**) axes. **d**, Odd (blue, left axis) and even (red, right axis) harmonic intensities as a function of the ellipticity of the driving field, for harmonics (left to right): H14–H15, H18–H19 and H21–H22. **e**, The reconstructed complex Berry phase, $\Delta\gamma_B$, as a function of the ellipticity of the driving field for each pair of neighbouring harmonics (that are presented above, in d). We resolve both the real (green, left axis) and imaginary (cyan, right axis) components of the Berry phase.

both the odd and even harmonics show fringe-like patterns oscillating out of phase with each other (Fig. 2d).

The oscillations of the HHG signal reveal the interferometric nature of the measurement. Both the odd and even harmonics result from the interference of the signals generated during two consecutive laser half cycles[32,33]. This interference encodes the relative phase accumulated between the two closed quantum paths that the system takes during successive half cycles. To retrieve the phase, we perturb the interferometric measurement by making the driving field weakly elliptic. We can then expand the geometric phase equation (1) to the first order in the ellipticity of the field, $\epsilon$ (Supplementary Information), leading to the accumulation of symmetric $\Delta\varepsilon_g(\epsilon)$ and anti-symmetric $\Delta\gamma_B(\epsilon)$ components. In the presence of C2 symmetry, as is the case for the Γ−K direction (or for any inversion-symmetric system), the perturbation is dominated by $\Delta\varepsilon_g$, which is symmetric along the two interferometer arms. Rotating the crystal off this axis gives rise to the anti-symmetric contribution, $\Delta\gamma_B(\epsilon)$, which has an opposite sign along the two sub-cycles. Note that the light-driven geometric phase also includes an imaginary part, which captures the quantum nature of the interaction and is associated with the contribution of the electron tunnelling across the band gap.

Finally, the complex perturbation is mapped onto the odd and even harmonics according to (Supplementary Information):

$$I_{odd,N}(\epsilon) \propto e^{-2\mathrm{Im}(\Delta\varepsilon_g(\epsilon))}|(\mathbf{E}^+_{0,N}e^{i\Delta\gamma_B(\epsilon)} + \mathbf{E}^-_{0,N}e^{-i\Delta\gamma_B(\epsilon)})|^2$$
$$I_{even,N}(\epsilon) \propto e^{-2\mathrm{Im}(\Delta\varepsilon_g(\epsilon))}|(\mathbf{E}^+_{0,N}e^{i\Delta\gamma_B(\epsilon)} - \mathbf{E}^-_{0,N}e^{-i\Delta\gamma_B(\epsilon)})|^2$$

(2)

where $\mathbf{E}^\pm_{0,N}$ are the unperturbed prefactors, containing the dipole couplings, corresponding to the interaction induced along positive and negative half cycles. As the measurement resolves the harmonic intensity, only the imaginary component of the symmetric part of the perturbation $\Delta\varepsilon_g(\epsilon)$ contributes, representing the suppression of the recombination probability with ellipticity. By contrast, the anti-symmetric phase $\Delta\gamma_B(\epsilon)$ can be directly observed, inducing clear oscillations between the neighbouring odd and even harmonics. Along the

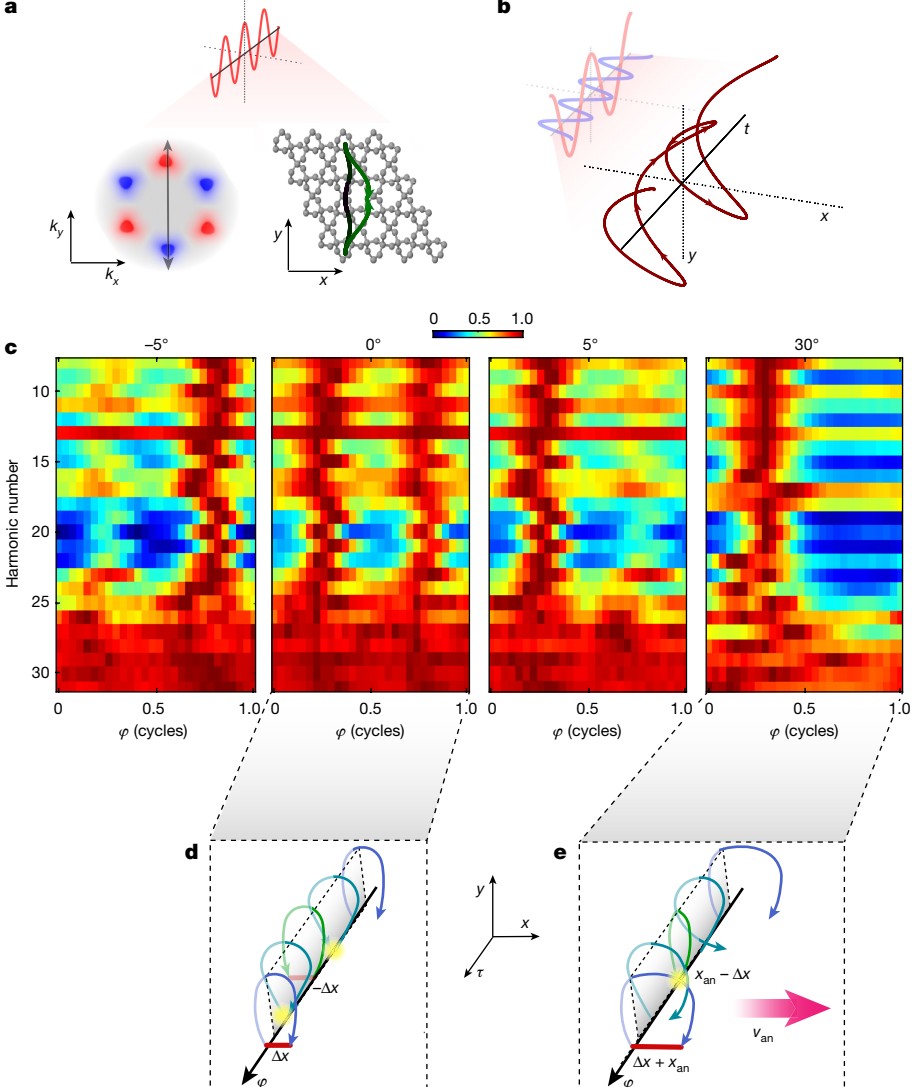

**Fig. 3 | Resolving the Berry curvature. a**, Intraband wavepacket evolution, in momentum (left) and real (right) space, induced by a linearly polarized field (dark green to light green, representing the evolution in time from the tunnelling time to the recombination time). The trajectory in the top and bottom planes corresponds to the positive and negative half cycles of the driving field, respectively. **b**, Time evolution of the two-colour orthogonally polarized field. The fundamental field is polarized along the $y$-direction and its second harmonic field along the $x$-direction, for a case of zero subcycle delay ($\varphi = 0$) between the two fields. **c**, The harmonics signal as a function of the two-colour delay ($\varphi$), resolved for different crystal orientations (left to right: −5°, 0°, 5° and 30°). Each row in the 2D plot is normalized by its maximal value. **d**,**e**, Schematic of real-space wavepacket evolution, during the fundamental first subcycle, for different time delays between the two fields. The lateral displacement of the electron–hole wavepacket at the recombination time ($\Delta x$) is induced by the two-colour field, in which the maximal and minimal harmonic signals are achieved at the minimum and maximum displacement, respectively (yellow marker). The lateral shift induced by the anomalous velocity is represented by $x_{an}$.

Γ−K axis, in which the interaction is dictated by the C2 symmetry, $\Delta\gamma_B(\epsilon) = 0$, and the interference is dominated by only $\Delta\varepsilon_g$ (Supplementary Information). We find that along this axis the even and odd harmonics show a similar response, decaying with the increasing ellipticity. Resolving the interference along the Γ−M axis gives access to $\Delta\gamma_B(\epsilon)$. As shown in equation (2), in this case we find the opposite response of the odd and even harmonics with $\epsilon$. This response serves as a sensitive probe of this phase, enabling its reconstruction.

Figure 2e presents the retrieved Berry phase as a function of the ellipticity of the driving field for harmonics H14–H15, H18–H19 and H21–H22. The reconstructed Berry phase increases with $\epsilon$, following the larger asymmetry induced by the elliptically polarized field (see Supplementary Information for a detailed description). Our reconstruction procedure is most accurate at lower ellipticity values. Moreover, for higher harmonics, the reconstructed Berry phase is larger, reflecting longer trajectories associated with these harmonics. The increase of the imaginary Berry phase captures the quantum nature of the interaction, originating from both the tunnelling mechanism and the reduced electron–hole overlap with increasing ellipticity. To the best of our knowledge, this is the first experimental observation and reconstruction of the interband Berry phase in crystals, resolved by strong-field light–matter interactions.

## Resolving the Berry curvature

Next, we extend our interferometry scheme to probe the well-known local geometrical property, the intraband Berry curvature ($\mathbf{\Omega} = \nabla_k \times \mathcal{A}$). Although the Berry phase and the Berry curvature are strongly related, their physical properties are inherently distinct, leading to different observations (Supplementary Information). The intraband Berry curvature gives rise to a large variety of phenomena, such as Hall

conductivity and orbital magnetism[4]. In particular, the application of an electric field induces a transverse current, normal to the Berry curvature direction, associated with the anomalous velocity ($\mathbf{v}_{an} = -\frac{e}{\hbar}\mathbf{F} \times \mathbf{\Omega}$, where $\mathbf{F}$ is the electric field). The anomalous velocity induces a drift of an electron trajectory in the lateral direction in the coordinate space (Fig. 3a), whereas its $k$-space trajectory remains unchanged. The HHG mechanism serves as an extremely accurate probe of the Berry curvature because of the lateral drift it induces[14]. This drift suppresses the spatial overlap between the electron and the hole, suppressing the recombination probability and therefore the HHG signal (Supplementary Information). Previous studies resolved the Berry curvature using HHG polarimetry, dictated by all components of the interaction—the intraband evolution as well as the dipole couplings[24–26]. By contrast, our measurement scheme provides a direct local probe of the anomalous velocity, isolating its impact on the light-driven trajectories.

In a broken-inversion-symmetry crystal, the Berry curvature inverts its sign between $\mathbf{k}$ and $-\mathbf{k}$. Therefore, when the interaction is driven by a single-colour field, an identical lateral drift is induced between the two consecutive half cycles, generating two mirror-imaged real-space paths (Fig. 3a). We show the role of the Berry curvature by driving the interaction with a laser field that holds the same mirror-symmetry property. This symmetry is achieved by combining the fundamental field with its orthogonally polarized second harmonic[34]. The total vector potential rotates in a 2D plane (Fig. 3b), inducing two mirror-imaged trajectories. This manipulation modifies the evolution of the wavefunction in a controllable manner—enhancing or compensating the anomalous velocity and the associated drift. Controlling the delay between the two fields, $\varphi$, shapes the instantaneous 2D laser field, driving the electron along or against the anomalous velocity direction. This manipulation suppresses or enhances the recombination probability and is directly mapped onto the HHG signal. Importantly, owing to symmetry, the $k$-space interferometer is balanced, in which the relative interband Berry phase cancels out. This scheme enables us to isolate the role of the Berry curvature and capture its direct influence on the electron trajectories.

Figure 3 experimentally resolves the role of the intraband Berry curvature on laser-driven electron trajectories. Figure 3c presents the harmonic signal as a function of the two-colour delay, $\varphi$, measured at different crystal orientations[24]. First, we focus on the Γ−K direction (0°), having a C2 symmetry, in which the Berry curvature is zero. When the interaction is driven by a single-colour field, the electron–hole follows a one-dimensional trajectory. The addition of a weak second harmonic field induces 2D mirror-symmetric trajectories having a lateral shift (Fig. 3d, red line, Δx), in which the interaction cannot distinguish between the left- or right-lateral displacement. This case is equivalent to an inversion-symmetric crystal, in which zero lateral shift is obtained twice within one period of the second harmonic field. Figure 3c shows this periodicity, identifying the fundamental symmetry of the interaction and the absence of the Berry curvature. A subtle rotation of the crystal, by just 5°, dramatically changes this observation, reducing the periodicity of oscillations to be a full cycle of the second harmonic field. Once the Berry curvature becomes non-zero, a small displacement is induced due to the anomalous velocity (Fig. 3e, $x_{an}$), driving the electrons along 2D mirror-symmetric paths. Here the addition of the second harmonic field identifies the role of the anomalous velocity, compensating or increasing the induced lateral drift. In this case, the total drift along the right or left direction becomes distinguishable. We maximize the symmetry breaking by rotating the crystal along the Γ−M axis (30°), maximizing the Berry curvature itself. In this case, there is only one delay in which both contributions—the anomalous velocity and the second harmonic field—compensate each other, reducing the periodicity of the measurement to be one second harmonic period. These results identify unequivocally the dominant role of the Berry curvature in the evolution of strong-field-driven electrons.

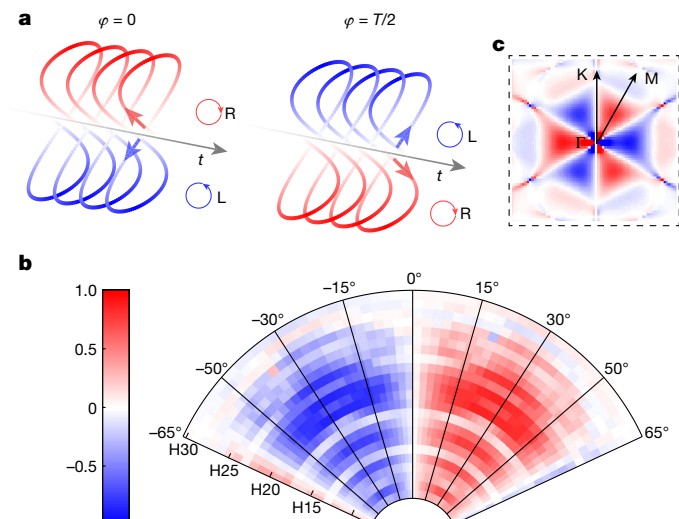

**Fig. 4 | Circular dichroism HHG spectroscopy. a**, The instantaneous chirality of the two-colour field, controlled by the two-colour delay configuration, reversing its sign between consecutive half cycles. **b**, Experimentally resolved HHG circular dichroism, $CD_{HHG}$, anti-symmetrized along 0° orientation (Supplementary Information), as a function of harmonic order (radial axis) and crystal orientation (azimuthal axis). **c**, The DFT-calculated Berry curvature of one of the conduction bands, $\Omega_{16}$, representing the origin of the HHG circular dichorism, dictating its symmetry properties.

The two-colour HHG scheme forms a unique configuration, enabling the detection of the Berry curvature in a time-reversal symmetric system. The high sensitivity is provided by the highly nonlinear nature of the interaction; the response during the first half cycle is localized around Γ−M (positive Berry curvature) and during the second half cycle around Γ−M′ (negative Berry curvature). Importantly, although the overall laser field is not chiral, the instantaneous chirality of the field changes its direction between two consecutive half cycles[35] (Fig. 4a). As we shift the two-colour delay by $T/2$ ($T$ is the second harmonic period), we reverse the instantaneous chirality. Therefore, the signal difference between these two delays reflects a circular dichroism (CD) measurement: $CD_{HHG} \equiv \frac{I(\varphi) - I\left(\varphi + \frac{T}{2}\right)}{I(\varphi) + I\left(\varphi + \frac{T}{2}\right)}$. In Fig. 4b, we plot the circular dichroism signal, resolved for different harmonic numbers, as a function of the orientation of the crystal. As can be observed, along Γ−K (0°) the circular dichroism signal vanishes, having its largest values around the maximal Berry curvature (Γ−M, 30°). Moreover, in contrast to the well-known linear optical schemes, the circular dichroism signal measured by this scheme is extremely high, approaching 70%. The high sensitivity is provided by the strong-field nature of the interaction, reflecting its exponential dependence on the Berry curvature (Supplementary Information).

In summary, our study presents a previously unknown formalism of the Berry phase, accumulated in both discrete and continuous space. HHG spectroscopy enables us to realize Berry-phase interferometry and probe the coherent properties of electron–hole wavefunction on a subcycle time scale. We experimentally demonstrate this scheme and resolve the generalized Berry phase across a large energy range. Extension of the approach to a two-colour field enables sensitive probing of the Berry curvature. The ability to resolve angstrom-scale displacement of the electron enhances the sensitivity of the measurement by orders of magnitude, enabling us to probe extremely low values of the curvature. We believe that the fundamental properties of our measurement will position HHG spectroscopy as a unique experimental scheme to identify Berry curvature and topological phases at higher conduction bands[36]. Importantly, this scheme provides opportunities

for Berry curvature measurements in insulators, probing a large range of condensed matter systems that cannot be resolved using transport measurements[4] or other techniques. Finally, our scheme opens new paths in probing light-driven band structure, in which the fundamental properties of the solid change during less than one optical cycle[37], exhibiting attosecond-scale topological phenomena[25,38].

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

## Data availability

The data and datasets that support the plots in this paper and other findings of this study are available from the corresponding authors upon reasonable request.

## Code availability

The custom code used for this study has been described in previous publications, and parts of it can be made available from the corresponding authors on reasonable request.

**Acknowledgements** We acknowledge S. Beaulieu, H. Beidenkopf and Y. Mairesse for their discussions and scientific advice. We thank E. Berg for his contribution to this work. N.D. is the incumbent of the Robin Chemers Neustein Professorial Chair. N.D. acknowledges the Minerva Foundation, the Israeli Science Foundation and the European Research Council for the financial support. A.J.U.-N. acknowledges financial support from the Rothschild Foundation and the Zuckerman Foundation. M.I. acknowledges funding of the DFG QUTIF grant IV152/6-2. Á.J.-G. acknowledges funding from the Horizon 2020 research and innovation programme of the European Union under the Marie Skłodowska–Curie grant agreement no. 101028938 and from the Comunidad de Madrid through TALENTO Grant 2022-T1/IND-24102. Á.J.-G. and M.I. acknowledge funding from the Horizon 2020 research and innovation programme of the European Union under grant agreement no. 899794. R.E.F.S. acknowledges support from the fellowship LCF/BQ/PR21/11840008 from the La Caixa Foundation (ID 100010434) and from the Horizon 2020 research and innovation programme of the European Union under the Marie Skłodowska–Curie grant agreement no. 847648.

**Author contributions** N.D. and M.I. supervised the study. A.J.U.-N. and L.F. conceived and planned the experiments. J.X. and B.Y. performed the DFT calculations. V.N., O.S. and M.I. developed the theoretical model. G.G.B., Á.J.-G. and R.E.F.S. performed the theoretical study and the numerical analysis. A.J.U.-N., T.A.-P., L.F. and B.D.B. performed the measurements. A.J.U.-N., L.F., S.S. and O.K. analysed the data. All authors discussed the results and contributed to writing the paper.

**Competing interests** The authors declare no competing interests.

**Additional information**
**Correspondence and requests for materials** should be addressed to Ayelet J. Uzan-Narovlansky or Nirit Dudovich.
