## [Peer Review File · Nature]

Manuscript Title: Observation of interband Berry phase in laser driven crystals

Reviewer Comments & Author Rebuttals

Reviewer Reports on the Initial Version:

Referees' comments:

Referee #1 (Remarks to the Author):

The paper entitled "Observation of interband Berry phase in laser driven Crystal" by Ayelet J. U.-N. et al., present a novel method to extract berry phase differences via attosecond interferometry.

"The manuscript shows a conceptually new formalism of the Berry phase, accumulated in both discrete and continuous space. HHG spectroscopy allows us to realize Berry phase interferometry and probe the coherent properties of electron-hole wavefunction on a sub cycle time scale. The authors experimentally demonstrate this scheme and resolve, for the first time, the generalized Berry phase across a large energy range. Extension of the approach to a two color field enables sensitive probing of the Berry curvature."

These experimental and theoretical results are impressive. I think the paper is well presented and encouraging. However, I have a few concerns before recommend it for publication.

Usually the Berry phase is defined in a closed path, however the interband Berry phase does not seem to follow a closed loop linear integral. Can the author comments on it, please?

I don't understand Eq. 2 very well, could the authors provide a more detailed explanation where is this Eq. 2 coming from and its relation with the Circular Dichroism?

At the beginning once we read the paper, we are thinking in a topological material, but alpha-quartz is not, the author should mention it clearly.

After this two minor points are fixed, I will recommend publication at Nature.

Referee #2 (Remarks to the Author):

The present work uses high harmonic spectroscopy to explore the Berry phase in dielectrics under optical excitation with intense laser fields. The authors refer to this as the 'interband Berry phase', a new term/quantity introduced here theoretically to account for both the excitation and recombination processes of electrons and holes. Thus, the authors consider it as a more 'generalized' quantity compared to the 'traditional' Berry phase,, which is primarily linked to the

intraband motion(s) of the charge carriers.

The authors argue for the importance of this new term in understanding topological phenomena in solids, assuming that the importance of the traditional 'intraband' Berry phase will naturally be inherited to the newly defined quantity.

Experimentally the study primarily involves control of the polarization state of the incoming laser pulse but experiments that utilize two-color, cross-polarized are also presented. The experimental work is systematic. There is generally a consistency among the theory behind the methodology presented and the experimental methods followed for the measurements. The manuscript is easy to read and to technically understand it.

This referee unfortunately cannot offer recommendation of this manuscript in its current form for publication in Nature for a number of reasons enlisted bellow:

1) By definition the new quantity to be measured, the interband berry phase, as the authors name it, is based on the bold assumption that high harmonic generation in quartz is an interband processes. More specifically, the authors argue along the lines of the generalized recollision picture introduced by Corkum and co-workers (Vampa et al., Nature 2015) as well as Brabec and coworkers (Vampa and Brabec PRL 2014) to explain early experiments of high harmonic generation in ZnO.

Why is this assumption then a bold one? Because this picture is unfortunately not supported by the experimental evidence in dielectrics—and more particularly in Quartz—the system of choice of the authors to present their first study on this topic.

Using attosecond spectroscopy (Garg et al, Nature 2016) has shown that the emission of high harmonics in this system is not compatible with the generalized recollision and interband picture. This was done by measuring high harmonic chirps and emission times—the hallmarks of the recollision picture. The group of Corkum, which proposed the generalized re-collision at the first place, has also transparently shown that this picture is not evidenced by their experiments in Quartz using a very different methodology to measure the emission phases (Hammond et al., Nature Photonics 2017). Moreover experiments for instance (Garg et al, Nat. Phot. 2018) and others have added more evidence for the inadequacy of the recollision picture in this system.

It is therefore not accidental that the first works on the Berry phase by T.T. Luu et al., Nature communications 2018 and H. Liu, Nat. Phys. 2017 have based their studies on the intraband picture.

I feel that these previous papers are improperly cited in the manuscript. These works have to be discussed at the very beginning of this manuscript while this manuscript need to start its story by quoting what will be new and more complete with the new methodology.

Obviously we need to be open to potential scientific evidence that contradicts previous works- this all what science is about. Yet this evidence here, to my regret, is very limited. The authors calculate a DFT bandstructure and contrasted it to a harmonic spectrum. They argue on a correlation between

amplitude of harmonics and the corresponding bands to make a case? But is this how nonlinear optics works? One band at a specific energy defines the harmonic amplitude at that energy alone? I feel that this argument is not helpful and I would encourage the authors to avoid it in potential next steps of this work. It is not even a simulation, but a visual comparison.

II) The authors argue on the importance of this work with reference to the importance of the Berry phase (the original one). But what they present is not the Berry phase, is something new. Previous works, (for example, Luu et al., Nature communications 2018) have used precisely the same system and it seems that their reconstructed Berry phase matches pretty well the expectations (to the extend a comparison is accurate). There the measured quantity is indeed connected directly to the broadly known Berry phase.

For this work to stand higher than the previous two papers on this topic, this referee would expect that the Berry phases measured would have a more important metrological value, benchmarked by theory of SiO₂ or, alternatively, other important experimental/theoretical quantities not captured by the previous works. I feel that this aspect is missing here and irrespectively of (I) I would feel uncomfortable to support this work without seeing these facts transparently presented.

III) More technically. The authors assume that the “excitation” matrix element is independent of the polarization state of the strong pulse. In this sense the polarized field is simply decomposed to two perpendicular fields that guide the motion of the excited carrier(s) only. Is there experimental evidence that this is the case? Alternatively, is it supported by rigorous calculations?

IV) It would be more appropriate and highly desirable to have simulations on the same system that is being studied. The authors show calculations in graphene? arguing that this still supports their argument. I believe that introducing new quantities requires dedicated simulations for the system in question to allow a strong case.

Referee #3 (Remarks to the Author):

Uzan-Narovlansky et al. present an interesting set of data by which they retrieve the Berry phase observed in HHG spectra by near-infrared driving of the coherent inter and intraband dynamics in a quartz crystal. To this end, they measure the emitted HHG in the multi-eV spectral range and perform interferometry while controlling the electrons' trajectories by the polarization state of the incident near-infrared driving field. The main claims of the manuscript are a new concept to investigate the Berry phase, in particular during the energy-discontinuous quantum evolution given by the interband polarization, and the first direct observation of the interband Berry phase. The data are clear, the analysis is sound, and the results are exciting. However, as spectacular as the results may seem assuming a first observation, there is evident overlap with existing works. Schmid et al., cited as [40] by the authors, have already presented direct measurement of the Berry phase in a topological insulator at lower frequencies from mid-IR to VIS/UV. It is unclear why this reference is somewhat neglected by placing it at the very end of the main text rather than discussing the similarity with the present data. To be clear: the differences are interband as well as intraband Berry phases on the one hand and intraband-only Berry phases on the other, while the similarities include

ballistic lightwave acceleration, high-harmonics generation, HHG interference, Berry curvature in general, the Berry curvature shown in Fig. 4c and [40] in particular, subcycle dynamics, polarization effects in HHG, Bloch's acceleration theorem, solid-state all-coherent polarization dynamics, and more. In this light, the authors would have to argue why their work is not an extension of previous works, that merely adds, technicalities aside, the interband Berry connection (although that result would still be quite interesting).

While I like the results, I must ask for their placement to be revised. Under the current circumstances, I would not support publication in Nature owing to the limited degree of novelty as described above. Moreover, I list a few more issues below which the authors can certainly address.

Validity of equation 1 is not straight-forward or intuitive to see. Some terms connect valence and conduction bands, others are intraband, without an easily recognizable pattern. The authors should discuss the structure more in detail.

The presentation of the data in Fig. 2 is not clear. The dashed lines suggest magnifications of areas of the spectra of panel c, but they actually show two cuts at the discrete harmonics. Moreover, panel e shows curves in the background which seem like the log of the intensity. No corresponding description is given. Then, the phase is shown by means of the real and imaginary part of $\Delta\gamma_B$, but for which harmonic? It would make sense to analyze the even harmonics, which also seems to be the case.

Also, since the distinction of odd and even harmonics is crucial, the spectra in b should be labelled accordingly (rather than just energy in eV, which almost coincides with the harmonic number, somewhat by accident, since $1200 \text{ nm} \sim 1 \text{ eV}$, the driving field energy).

Somewhat disappointingly, the color plot of Fig. 3c does not have a scale bar. It is thus, strictly speaking, worthless, as there could be any mapping of color and intensity.

One of the early investigations of this field which established the mechanism of coherent interband excitation and intraband transport is missing (Schubert et al., Nature Photonics 8, 119 (2014)).

Author Rebuttals to Initial Comments:

1 Referee 1

The paper entitled “Observation of interband Berry phase in laser driven Crystal” by Ayelet J. U.-N. et al., present a novel method to extract berry phase differences via attosecond interferometry: “The manuscript shows a conceptually new formalism of the Berry phase, accumulated in both discrete and continuous space. HHG spectroscopy allows us to realize Berry phase interferometry and probe the coherent properties of electron-hole wavefunction on a sub cycle time scale. The authors experimentally demonstrate this scheme and resolve, for the first time, the generalized Berry phase across a large energy range. Extension of the approach to a two color field enables sensitive probing of the Berry curvature.”

These experimental and theoretical results are impressive. I think the paper is well presented and encouraging. However, I have a few concerns before recommend it for publication.

We thank the referee for his careful reading, valuable comments and support in the acceptance of the paper to Nature. In the following we provide a point-by-point response to the comments raised by the referee.

Comment: Usually the Berry phase is defined in a closed path, however the interband Berry phase does not seem to follow a closed loop linear integral. Can the author comments on it, please?

Answer: Indeed, the Berry phase, as any geometrical phase is a physical observable only if the wavefunction follows a close loop in the parameter space. Different geometrical phases distinct from each other by their parameter space and their adiabatic conditions. However, all classes of geometrical phases require closed loop configurations. The common case of the Berry phase in condensed matter defines the parameter space as the momentum space, where a closed loop is achieved by integrating over the periodic BZ (figure 1a in the main text, left panel). In our work, the wavepacket evolves along a closed loop as well, however, in contrast with the common case, here the parameter space is momentum-energy space (figure 1a in the main text, right panel). The closed loop is dictated by the nature of the interband HHG process – where the loop is closed by the recombination of the electron and the hole.

Changes introduced: To clarify this point, we modified line 95 in the main text, which now reads: ” First, the interaction is driven in a closed loop in energy-momentum space, induced by the ionization and the recombination steps, leading to the accumulation of the interband Berry phase [1]”

Comment: I don’t understand Eq. 2 very well, could the authors provide a more detailed explanation where is this Eq. 2 coming from and its relation with the Circular Dichroism?

Answer: We thank the referee for this important comment. Indeed, Eq. 2 was not well introduced. Following your comment, we have modified it in the revised version. In the following we provide a detailed description of its derivation. Equation 2 captures the interference between two electron trajectories, induced along positive and negative half cycles, which are modified

by the laser ellipticity (ϵ). It focuses on the regime of small ellipticities, $\epsilon \ll 1$, where the main parameters of the interaction are unchanged, while the ellipticity of the field induces a small lateral drift to the electron trajectory. We have expressed the semi-classical action as: $S = S_{inv} + \gamma_B$, where S_{inv} represents the inversion symmetric action and γ_B the non symmetric action, defined by the Berry phase. The modification of the semi-classical action with ϵ can be expressed as:

$$S^{(i)}[\mathbf{K}, t', t] \approx S_0^{(i)} + \frac{\partial S^{(i)}}{\partial \epsilon} \epsilon = S_{inv,0} + \gamma_{B,0}^{(i)} + \frac{\partial S_{inv}}{\partial \epsilon} \epsilon + \frac{\partial \gamma_B^{(i)}}{\partial \epsilon} \epsilon + \frac{\partial S_{inv}}{\partial \epsilon^2} \epsilon^2 \dots \quad (1)$$

Here K , t' , and t are the stationary crystal momentum, ionization and recombination times, $S_{inv,0}$ and $\gamma_{B,0}^{(i)}$ are the unperturbed symmetric and non-symmetric parts of the action, for linear polarized field. Importantly, the first derivative of S_{inv} with respect to ϵ vanishes due to inversion symmetry. Therefore, only the second order contributes: $\Delta S_{inv} \equiv \Delta \varepsilon_g \propto \frac{\partial S_{inv}}{\partial^2 K_{st}^2} \epsilon^2$. This term represents the suppression of the recombination due to the lateral shift of the wavepacket, induced by the laser ellipticity. Such suppression is well known in the gas phase case [2]. Since the ellipticity modifies the evolution of the wavepacket in energy-momentum space, it also leads to the modification in the Berry phase defined by the first-order perturbation term $\Delta \gamma_B = \frac{\partial \gamma_B^{(i)}}{\partial \epsilon} \epsilon$ (as this contribution is not inversion symmetric.)

The perturbation in the semiclassical action modifies the emitted field according to: $E \approx E_0 \exp(i\Delta \gamma_B + i\Delta \varepsilon_g)$, where E_0 represents the unperturbed field. The perturbation of two consecutive half-cycles is mapped onto the harmonics spectrum. Odd harmonics represent constructive interference between the fields emitted along the positive and negative half cycles, while even harmonics represent their destructive interference:

$$E_{even}^{odd}(\epsilon) = E^+(\epsilon) \pm E^-(\epsilon) \approx [E_0^+ \exp(i\Delta \gamma_B^+ + i\Delta \varepsilon_g^+) \pm E_0^- \exp(i\Delta \gamma_B^- + i\Delta \varepsilon_g^-)] \quad (2)$$

where $\Delta \gamma_B^\pm$, $\Delta \varepsilon_g^\pm$ represent the perturbation along the first and second half cycle. Along $\Gamma - M$, $\Delta \gamma_B^+ = -\Delta \gamma_B^- \equiv \Delta \gamma_B$, and $\Delta \varepsilon_g^+ = \Delta \varepsilon_g^- \equiv \Delta \varepsilon_g$. Therefore:

$$I_{even}^{odd}(\epsilon) \propto e^{-2\text{Im}(\Delta \varepsilon_g)} |E_0^+ \exp(i\Delta \gamma_B) \pm E_0^- \exp(-i\Delta \gamma_B)|^2$$

Changes introduced: Following the referee's comment, we introduced changes to the equation presented in the main text and also added the derivation above to the Supplementary Information (chapter 6.2). The text now reads:

Finally, the complex perturbation is mapped into the odd and even harmonics according to (see SI):

$$\begin{aligned} I_{odd}(\epsilon) &\propto e^{-2\text{Im}(\Delta \varepsilon_g(\epsilon))} |(E_0^+ e^{i\Delta \gamma_B(\epsilon)} + E_0^- e^{-i\Delta \gamma_B(\epsilon)})|^2 \\ I_{even}(\epsilon) &\propto e^{-2\text{Im}(\Delta \varepsilon_g(\epsilon))} |(E_0^+ e^{i\Delta \gamma_B(\epsilon)} - E_0^- e^{-i\Delta \gamma_B(\epsilon)})|^2 \end{aligned} \quad (3)$$

where E_0^\pm are the unperturbed prefactors, containing the dipole couplings, corresponding to the interaction induced along positive and negative half cycles.

Comment: At the beginning once we read the paper, we are thinking in a topological material, but alpha-quartz is not, the author should mention it clearly.

Answer: We agree with the referee and therefore we have modified the text to explain the difference between the intraband Berry phase, often associated with topological phases, and the inter-band Berry phase we have introduced and measured in this work.

In our study we show that even in **trivial** insulators a laser field can drive the system along a closed loop in the generalized space of crystal momentum (a continuous variable) and band index (a discrete variable). The phase along this loop is also gauge invariant, it reflects the intrinsic properties of the system (the Berry connections along the loop), and is observable, e.g. via a nonlinear optical response as we have demonstrated.

Changes introduced: We added a short discussion in the main text (line 99) which clarifies the main differences between previous studies and our study: "The short time scale of the mechanism provides the key advantage of our scheme. Schmid et al. Ref [16] revealed the Berry curvature of topological insulators via HHG driven by THz field, having a fundamental period of 40 fs. In their study, topology enables to overcome dephasing and scattering mechanisms, revealing the geometrical properties of the system. Our measurement, performed on an attosecond time scale, allows the probing of the Berry phase in trivial insulator."

After this two minor points are fixed, I will recommend publication at Nature.

2 Referee 2

Comments I-III: By definition the new quantity to be measured, the interband berry phase, as the authors name it, is based on the bold assumption that high harmonic generation in quartz is an interband processes. More specifically, the authors argue along the lines of the generalized recollision picture introduced by Corkum and co-workers (Vampa et al., Nature 2015) as well as Brabec and coworkers (Vampa and Brabec PRL 2014) to explain early experiments of high harmonic generation in ZnO.

Why is this assumption then a bold one? Because this picture is unfortunately not supported by the experimental evidence in dielectrics—and more particularly in quartz—the system of choice of the authors to present their first study on this topic.

Using attosecond spectroscopy (Garg et al, Nature 2016) has shown that the emission of high harmonics in this system is not compatible with the generalized recollision and interband picture. This was done by measuring high harmonic chirps and emission times—the hallmarks of the recollision picture. The group of Corkum, which proposed the generalized re-collision at the first place, has also transparently shown that this picture is not evidenced by their experiments in quartz using a very different methodology to measure the emission phases (Hammond et al., Nature Photonics 2017). Moreover experiments for instance (Garg et al, Nat. Phot. 2018) and others have added more evidence for the inadequacy of the recollision picture in this system.

It is therefore not accidental that the first works on the Berry phase by T.T. Luu et al., Nature communications 2018 and H. Liu, Nat. Phys. 2017 have based their studies on the intraband picture.

I feel that these previous papers are improperly cited in the manuscript. These works have to be discussed at the very beginning of this manuscript while this manuscript need to start its story by quoting what will be new and more complete with the new methodology. Obviously we need to be open to potential scientific evidence that contradicts previous works- this all what science is about. Yet this evidence here, to my regret, is very limited. The authors calculate a DFT band structure and contrasted it to a harmonic spectrum. They argue on a correlation between amplitude of harmonics and the corresponding bands to make a case? But is this how nonlinear optics works ? One band at a specific energy defines the harmonic amplitude at that energy alone? I feel that this argument is not helpful and I would encourage the authors to avoid it in potential next steps of this work. It is not even a simulation, but a visual comparison.

Comment IV) It would be more appropriate and highly desirable to have simulations on the same system that is being studied. The authors show calculations in graphene? arguing that this still supports their argument. I believe that introducing new quantities requires dedicated simulations for the system in question to allow a strong case.

Answer: We really appreciate these comments by the referee and fully agree with the general point – the ability to induce an internal interferometer and resolve the Berry phase is based on a complete understanding of the underlying physical mechanism. We also agree with

the referee that our original manuscript did not provide sufficient evidence of the key physical mechanism underlying our observations, as well as its proper comparison to the previous works.

The two major complementary mechanisms responsible for high harmonic generation in solids are associated with intraband currents [3–5] and interband polarization [6]. The balance between them depends on the laser parameters (carrier frequency, intensity and pulse duration), the properties of the medium, and can change across the harmonic spectrum.

Following the referee’s criticism, over the past year we have performed an entirely new experimental study that resolves the origin of high harmonic emission in quartz in our experiments, showing the dominant role of the interband mechanism under our experimental conditions. In addition, we have performed systematic theoretical analysis, including full first-principles microscopic calculations in quartz and also simplified modeling of the macroscopic far-field response. Our theoretical results agree with different experimental observations [7–9] and show how the dominant harmonic generation mechanism varies with different experimental conditions. Below we provide a detailed description of our new experimental and theoretical study, starting with theory.

2.1 First-principle simulations

We performed time-dependent simulations in alpha quartz, using the code Quantum Espresso with a PBE functional on a 12x12x12 Monkhorst-Pack grid for computing the band structure. We then used the Wannier90 code [10] to generate the representation of the field-free Hamiltonian within the basis of maximally-localized Wannier functions. This procedure allows one to obtain a consistent phase relation between the complex-valued dipole couplings at different crystal momenta. The Bloch states were projected onto the sp_3 orbitals of Si and the p orbitals of oxygen, totalling 30 bands. The α -quartz real-space, reciprocal space, and band structures are shown in Fig. 1. The Hamiltonian in the basis constructed from the Wannier functions was then propagated in the presence of the laser electric field using the density matrix formalism and the code we have described in ref. [11].

The laser wavelength was set to 1200nm, the full-width at half maximum pulse duration was set to 50 fs, with the pulse intensity varied in the range of 0.9-30 TW/cm². The dephasing time was set to one or two cycles of the driving field, $T_2 = T_0 = 2\pi/\omega$. Thanks to the shape of the Brillouin zone, the k_z component could be uncoupled in the calculation, allowing us to use fewer points along k_z and thus making the time-dependent calculations feasible within a reasonable amount of time.

We performed calculations for the carrier-envelope phase CEP=0 and CEP= π and averaged the results while imposing the symmetry constraints along each emission direction (currents are subtracted along $\Gamma - M$ and added along $\Gamma - K$). This does not affect symmetry-allowed harmonics, or the weights between intra- and inter-band components, which is our main interest, but removes the numerical artifacts that arise due to finite number of k-grid points, which

Figure 1: (a) α -quartz structure, (b) reciprocal space and (c) band structure. The unit cell is drawn in the center of the figure. The band structure was obtained with a PBE functional. Our Wannier90 interpolation projects over the p orbitals of the O atoms, and the sp_3 orbitals of the Si atoms. The HOMO energy is set to 0 eV. In the Wannierization procedure, we use a window up to 12 eV, ensuring optimal fit to the band structure within this window.

can lead to weak symmetry-forbidden harmonic lines. With this procedure, convergence was obtained for a 100x100x10 k-point grid.

The current operator is defined as

$$\hat{\mathbf{J}} = \frac{i|e|}{\hbar} [\hat{\mathbf{r}}, \hat{H}(\mathbf{k}, t)], \quad (4)$$

where $\hat{\mathbf{r}}$ is the position operator and $\hat{H}(\mathbf{k}, t)$ is the time-dependent Hamiltonian. In the length gauge, $H(\mathbf{k}, t) = H_0(\mathbf{k}) + |e| \mathbf{E}(t) \cdot \mathbf{r}$, so that the interaction term commutes with the position operator and the current operator can be written as

$$\hat{\mathbf{J}} = \frac{i|e|}{\hbar} [\hat{\mathbf{r}}, \hat{H}_0(\mathbf{k})]. \quad (5)$$

Following the definition of the position operator given by Blount [12] in terms of the Berry connection $A_{nn'}$,

$$\hat{\mathbf{r}} f_n(\mathbf{k}) = i \partial_{\mathbf{k}} f_n(\mathbf{k}) + \sum_{n'} A_{nn'}(\mathbf{k}) f_{n'}(\mathbf{k}), \quad (6)$$

we can write the current operator in a way such that we can separate intraband and interband currents,

$$\hat{\mathbf{J}} = -\frac{|e|}{\hbar} (\nabla_{\mathbf{k}} H_{0,nn}(\mathbf{k}) - i [\mathbf{A}(\mathbf{k}), H_0(\mathbf{k})]_{nn}). \quad (7)$$

The first term is non-zero when $n = m$ and is associated with the intraband contribution, while the second term contains the interband contribution.

Our results, presented in Figure 2, show that under the experimental conditions considered in our analysis, where the laser intensity approaches the experimental value, 30 TW/cm², the interband mechanism dominates. The same is found at lower intensities.

Figure 2 shows the results for the field linearly polarized along the $\Gamma - M$ direction, i.e., along the direction where the Berry curvature is maximal. We see that the interband current contribution is higher by two to three orders of magnitudes (Figures 2c and 2d) compared with the intraband contribution. We note that these results do not depend on the dephasing time, as shown in Figure 3.

We shall see below that these conclusions are consistent with available experiments for different laser parameters and by no means contradict the observations of intra-band mechanism at shorter wavelengths [7].

Our first-principle simulations predict that we should observe attochirp of high harmonics in our experimental conditions. This is the subject of our additional experimental measurements.

Figure 2: Calculated HHG spectrum for quartz along $\Gamma - M$, for the parallel (a,c) and perpendicular (b,d) components. The laser intensity is $I = 0.9 \text{ TW/cm}^2$ in panels (a,b) and $I = 30 \text{ TW/cm}^2$ in panels (c,d); the dephasing time is $T_2 = 2T_0$. The intra/interband currents are plotted in red/blue.

Figure 3: Comparison of the calculated HHG intensities for the perpendicular polarization component, for a laser intensity of $I = 0.9 \text{ TW/cm}^2$, for $T_2 = T_0$ (a) and $T_2 = 2T_0$ (b).

2.2 Experimental study of the HHG origin

We have performed a new experimental study that reveals the underlying HHG mechanism in our experiments. This experiment aims to confirm the atto-chirp of the emitted harmonics in the spectral region of interest. To this end, we probe the HHG mechanism in quartz via the application of the in-situ scheme [13], which we have developed during the past two decades, using it in a large variety of atomic and molecular systems to resolve a range of fundamental phenomena such as the tunneling dynamics [14, 15] or multielectron processes [16].

The in-situ scheme has now been extended to condensed matter systems. This scheme was first applied by the NRC group to reveal the role of the interband mechanism in a low band gap crystal such as ZnO [6]. Later, this scheme was applied by several groups to identify the interband process in a high bandgap system such as MgO [17, 18]. This observation was then confirmed by alternative spectroscopic approaches such as the CEP scans [9, 19] or the appearance of spectral caustics [17]. Recently, the Weizmann and the MBI teams have applied the in-situ scheme in MgO, and resolved laser-induced closing of the band-gap between adjacent conduction bands, revealing the mapping between the HHG and the different conduction bands [20].

Here, we apply the in-situ scheme in quartz. Our experiment consists of a strong IR field together with a weak second harmonic (SH) field, controlling their sub-cycle delay τ . Each trajectory acquires an additional complex phase $\sigma(\tau)$, which is accumulated along the entire trajectory, serving as a sensitive label of its temporal properties. We can describe this phase shift as a perturbation to the total action accumulated by the electron as: $S(\tau) = S_0 + \sigma(\tau)$. The real part, $Re(\sigma)$, is associated with an additional phase accumulated by the electron after injection (tunnelling), while the imaginary part, $Im(\sigma)$, is associated with a small perturbation of the injection (tunneling) process [14]. The additional phase is mapped into the intensity of both even and odd harmonics.

Scanning τ modifies $\sigma(\tau)$ in a periodic manner, modulating the harmonic spectrum. These modulations (i) reveal the symmetry of the solid-state medium and (ii) provide insight into the dynamical properties of the interaction.

First, we compare the in-situ measurement performed in MgO and quartz. Figure 4 presents the HHG spectrum as a function of τ for MgO and quartz. Clear modulations of the harmonic signal are observed for both systems. Before we carefully analyze the phase associated with each harmonic order, we focus on two main observables: the modulation's periodicity and the modulation's phase of both even and odd harmonics. In MgO the harmonics oscillate with $2\omega_{SH}$ frequency (ω_{SH} is the SH frequency), where even and odd harmonics are modulated out of phase. In strong contrast, in quartz, the signal oscillates with ω_{SH} frequency, where odd and even harmonics are modulated approximately in phase. What is the origin of this striking difference between the two systems?

We can understand this difference by considering the generation process induced during two consecutive half cycles. The exact expressions are derived in the new chapter in the SI (chapter

Figure 4: **Two-color HHG spectroscopy resolved in MgO and quartz crystals.** (a) and (b) Oscillating harmonic spectrum resolved in MgO and quartz crystals, respectively, as a function of the time delay between the two fields (presented with respect to the SH cycle period). (c) Comparison between the oscillation of the even (H18, red) and odd (H19, blue) harmonics, generated in MgO crystal, showing *out of phase* oscillations. (d) Comparison between the oscillation of the even (H20, red) and odd (H19, blue) harmonics, generated in quartz crystal, showing *in phase* oscillations.

5.3), here we present a general description that captures the complex nature of the perturbation. The harmonic fields emitted during each half cycle can be described as: $E_{1,2} = \alpha_{1,2}e^{i\varphi_{1,2}}$, where 1,2 represents the first and second half-cycles respectively. $\alpha_{1,2}$ represents the amplitude of the harmonic emission, while $\varphi_{1,2}$ represents its phase, which is dictated by the semi-classical action, S , as well as the dipole phase. Adding the SH field adds a complex perturbation to the harmonic's field emitted during the two half-cycles, represented by σ_1 and σ_2 . The HHG spectrum encodes the interference between the two emission bursts according to:

$$I^N \propto |\alpha_1 e^{i\varphi_1} e^{i\sigma_1(\tau)} \pm \alpha_2 e^{i\varphi_2} e^{i\sigma_2(\tau)}|^2 \quad (8)$$

where $+$ is for odd and $-$ is for even harmonics. In systems with inversion symmetry [13, 14]: $\alpha_1 = \alpha_2$, $\varphi_1 = \varphi_2$, and $\sigma_1 = -\sigma_2$, therefore:

$$I^N \propto |\alpha|^2 |e^{i\sigma_1(\tau)} \pm e^{i\sigma_2(\tau)}|^2 \propto \begin{cases} 1 + (\sigma^i)^2 - (\sigma^r)^2 & N \text{ is odd} \\ (\sigma^i)^2 + (\sigma^r)^2 & N \text{ is even} \end{cases} \quad (9)$$

Here $\sigma_{1,2}^i = \text{Im}(\sigma_{1,2})$ and $\sigma_{1,2}^r = \text{Re}(\sigma_{1,2})$.

Thus, in inversion symmetric systems, the first order vanishes and the perturbation appears in the **second** order only. Scanning the two-color delay thus modulates σ by ω_{SH} and the harmonic signal by $2\omega_{SH}$ oscillations. In contrast, in systems with broken inversion symmetry the response to the perturbation is completely different. In this case, the harmonic signal

behaves as:

$$I^N \propto \begin{cases} 2\alpha_1\alpha_2 \cos \Delta\varphi(1 + \sigma_1^i + \sigma_2^i) + 2|\alpha_1|^2\sigma_1^i + 2|\alpha_2|^2\sigma_2^i - 2\alpha_1\alpha_2 \sin \Delta\varphi(\sigma_1^r - \sigma_2^r) + O(\sigma^2) & N \text{ is odd} \\ -2\alpha_1\alpha_2 \cos \Delta\varphi(1 + \sigma_1^i + \sigma_2^i) + 2|\alpha_1|^2\sigma_1^i + 2|\alpha_2|^2\sigma_2^i + 2\alpha_1\alpha_2 \sin \Delta\varphi(\sigma_1^r - \sigma_2^r) + O(\sigma^2) & N \text{ is even} \end{cases} \quad (10)$$

where $\Delta\phi = \phi_1 - \phi_2$. Note that $\sigma(\tau)$ is a periodic function of the two-color delay τ . We see that the perturbation appears in the **first** order, leading to ω_{SH} oscillations.

This analysis fully captures our experimental observations of the different oscillation periods in MgO crystal vs. quartz crystal.

Next, we note that when the perturbation is dominated by its real component, the neighboring even and odd harmonics oscillate out of phase. In contrast, when the perturbation is dominated by the imaginary component, the odd and even harmonics will oscillate in phase [6, 14].

Clearly, the perturbation in MgO is dominated by the real component. In contrast, our experimental results reveal that the perturbation in quartz is dominated by the imaginary component of the perturbation. The dominant role of the imaginary perturbation originates from two important properties. The first is the high band gap of quartz (9.3eV), leading to crucial contribution of tunneling and therefore large imaginary action. The second is the asymmetry in tunneling during the two consecutive half cycles, associated with the symmetry breaking ($\sigma_1 \neq \sigma_2$) within each cycle.

Importantly, the tunneling step is common for both HHG mechanisms, inter- and intra-band. The modulation associated with this step does not distinguish between the two. The interplay of inter-band and intra-band mechanisms is encoded in the real perturbation associated with the phase accumulated by the trajectory. For the inter-band mechanism, this phase reflects the length of the trajectories and the mapping between their length and the harmonic orders.

We can isolate the oscillating real perturbation from the imaginary one by combining the pairs of neighboring harmonic orders:

$$I^{N_{odd}} - I^{N_{even}} \propto \alpha_1\alpha_2 \sin(\Delta\varphi)(\sigma_1^r - \sigma_2^r) \quad (11)$$

Such analysis removes the imaginary contribution and isolates the real perturbation.

Figure 5 presents the modulation phase, focusing on the harmonic emission associated with the first conduction band (HH11-HH17), for the three values of the fundamental field's intensity. This analysis shows that the modulations of even and odd harmonics are slightly out of phase, and that their phase difference changes with the harmonic order. In addition, the phase variation of both even and odd harmonics strongly depends on the fundamental field intensity, becoming almost flat at high laser intensities. We extract $I^{N_{odd}} - I^{N_{even}}$ and resolve

Figure 5: **Resolving the interband mechanism captured by the harmonics' oscillation phase.** (a-c) The oscillation phase of the even (I_{2N} , blue) and the odd (I_N , red) harmonics, as well as the oscillation of their difference ($I_N - I_{2N}$, black), generated using fundamental's field intensities of approximately $3.8 \cdot 10^{13} \text{ W/cm}^2$, $4.3 \cdot 10^{13} \text{ W/cm}^2$, $5 \cdot 10^{13} \text{ W/cm}^2$. (d) Semi-classical calculations of the recombination time as a function of the emitted photon energy, for different driving field amplitudes (light to dark: high to low laser intensity).

the modulation phase of each pair of neighboring harmonics. Such analysis reveals the temporal information recorded in the experimental results (equation 11).

Crucially, we find that the modulation phase changes significantly with the harmonic order, resolving the clear fingerprint of the interband mechanism. The variation of this phase from one harmonic pair to another originates from the recombination dynamics and identifies the mapping between the recombination time and harmonic order. Such response has been well established in gas phase [13] and, as the referee correctly points out, extended to condensed matter systems by Vampa et al in ZnO [6].

Importantly, our measurements also show how the interplay between intra-band and inter-band mechanisms changes with intensity. In our measurements, the spectral slope of the modulation phase decreases as we increase the fundamental field intensity. Such a response is in full agreement with the semi classical inter-band picture. At the same time, as we increase the laser intensity, the mapping between recombination time and harmonic order becomes more flat, and thus the attochirp reduces. Our experimental results point to the fact that if HHG is performed in quartz with very short driving pulses, which in practice necessitates high intensities, all harmonics up to H17 will be emitted in phase, consistent with the intra-band mechanism. However, in our experiment, this is clearly not the case.

2.3 Analysis of HHG mechanisms based on macroscopic simulations

We now complement our microscopic analysis with macroscopic simulations, albeit for a simplified microscopic model. This additional study allows us to better connect our numerical calculations with experimental results, since the latter deal with the far-field HHG signal. Crucially, this study also allows us to better examine the interplay between the different HHG mechanisms under different experimental conditions, providing a broader picture and connecting to previous experimental works of HHG in quartz, performed under different experimental conditions.

To this end, we have developed a novel approach based on real-space-resolved evaluation of the generated light in the basis of Wannier orbitals.

To demonstrate the strength and relevance of this method, let us start with the description of the HHG using the adiabatic Houston basis and write the time-dependent wavefunction with the initial crystal momentum k_0 as follows [21]:

$$|\psi(t)\rangle = a_{m,k_0}(t)|\psi_{m,k_0}^{(H)}(t)\rangle. \quad (12)$$

The time-evolution of the amplitude $a_{m,k_0}(t)$ is given by

$$i\frac{\partial a_{m,k_0}}{\partial t} = \sum_{m'} [\epsilon_{m'}(k_0 + A(t))\delta_{m,m'} - F(t)d_{m,m'}(k_0 + A(t))] a_{m',k_0}(t), \quad (13)$$

where $\epsilon_{m'}(k)$ and $d_{m,m'}(k)$ are the band m energy and the transition dipole moment between bands m and m' at crystal momentum k , and $F(t)$ and $A(t)$ are the external time-dependent electric field and vector potential at time t . Assuming negligible depletion of the ground state, the intra- and interband current calculated from this model are given as follows:

$$j^{(ra)}(t) = - \sum_m |a_{m,k_0}(t)|^2 \langle \psi_{m,k_0+A(t)} | \hat{p} | \psi_{m,k_0+A(t)} \rangle, \quad (14)$$

$$j^{(er)}(t) = - \sum_{m \neq m'} [a_{m,k_0}^*(t)a_{m',k_0}(t) \langle \psi_{m,k_0+A(t)} | \hat{p} | \psi_{m',k_0+A(t)} \rangle + c.c.]. \quad (15)$$

We now transform our representation into the Wannier basis, resulting in the following expression for the intraband emission spectrum [22]:

$$\tilde{j}_{\Delta_R}^{(ra)}(\Omega) = \sum_m \int_{-\infty}^{\infty} dt |a_{m,k_0}(t)|^2 e^{-i(k_0+A(t))\Delta_R} p_{m,m}^{\Delta_R} e^{i\Omega t}. \quad (16)$$

Here Δ_R denotes the spatial separation between the two Wannier states (see chapter 3.3 SI). If we now analyze the integral over t in Eq. (16) using the saddle-point approximation, we obtain the following saddle-point condition [22]:

$$0 = \Omega - F(t_s)\Delta_R \implies N \propto \Delta_R, \quad (17)$$

Thus, the intraband HHG emission above the minimum band gap will be associated with lattice site separations that satisfy Eq. (17).

In contrast, the interband emission in Eq. (14) depends on the interband momentum matrix element, which rapidly decays with lattice site separation. Consequently, the interband HHG emission will be dominated by coherences between relatively close lattice sites.

Therefore, we conclude that the intra- and interband HHG are governed by distinct real-space dynamics, and that resolving the HHG emission above the minimum band gap according to lattice site separation provides a means for distinguishing between the two mechanisms.

We now use this insight to analyse the combination of microscopic and macroscopic simulations. To make the macroscopic simulations feasible, we use a simple one-dimensional microscopic model of a crystal, with the periodic potential adjusted to obtain the desired band gap, see SI chapter 3 for the full description of the model.

We simulate HHG emission across a radially symmetric Gaussian beam-front with a beam-waist of 100 μm , peak electric field amplitude 0.014 a.u., and a pulse duration of eight optical cycles for a driving field wavelength of 1200 nm. Figure 6 depicts the lattice site separation-resolved radially integrated HHG spectra, after propagating the near-field signal from a 2D sample by a distance of 1 m into the far field, assuming 10 fs dephasing time. The dashed vertical red lines denote the harmonic order corresponding to the minimum band gap. The dashed white line depicts the minimum lattice site separation for the intraband emission at a given harmonic order, calculated using Eq. (17). The changing interplay of the two mechanisms across the harmonic spectrum is clear, with the highest harmonics dominated by the inter-band mechanism and the lower to mid-range harmonics demonstrating comparable contributions from both.

We are now able to study the relative importance of the two HHG mechanisms on the driving field wavelength. Figure 7 presents the HHG spectrum as a function of lattice site separation for wavelengths of 0.6 μm , 0.8 μm , 1.0 μm , and 1.2 μm . For longer driving field wavelength (c and d), the intra- and interband components of the spectrum are clearly separated above the minimum band gap, where the emission is clearly dominated by the near-zero lattice separation emission and is predominantly interband. For shorter wavelengths (panels (a) and (b)) the situation is different. Reducing the wavelength increases the strength of the intraband emission, and the separation between intra- and interband emission in the total HHG spectrum is less clear. At a shorter wavelength, 0.6 μm (a), the intraband emission is the dominant HHG mechanism.

This conclusion is supported by the time-frequency Gabor analysis of the emission shown in Fig. 8, for the same driving laser wavelengths of (a) 0.6 μm , (b) 0.8 μm , (c) 1.0 μm , and (d) 1.2 μm . All Gabor transforms are normalized according to their respective maxima for photon energies above the minimum band gap. For the driving laser wavelength of 0.6 μm in (a), where the HHG emission is dominated by the intraband mechanism, no attochirp is observed,

Figure 6: Lattice site separation-resolved far-field HHG spectra calculated with driving laser wavelengths of 1200 nm, driving field beam waist of $\sigma = 100 \mu\text{m}$ and a driving electric field amplitude of 0.014 a.u.. A far-field propagation distance of 1 m is considered. The dashed white lines denote the minimum lattice site separation separation for HHG emission at a given harmonic order calculated using Eq. (17) and the vertical dashed red lines denote the harmonic order of the minimum band gap.

as expected.

As we go to longer wavelengths, the attochirp becomes more pronounced. The above-band-gap emission calculated with a 1.2 μm driving laser wavelength shown in (d) exhibits a clear attochirp for nearly all photon energies above the minimum band gap, showing the dominant role of the interband emission. The harmonics emission in the spectral range of HH14-HH21, used for analysis in our experiment, clearly lie within the energy region which exhibits a clear attochirp.

Our results clearly show that the dominant mechanism of HHG emission shifts from intra- to interband as the driving laser wavelength is increased and that interband emission is the dominant HHG mechanism for the experimental conditions used in this work.

At the same time, our results show that experiments performed with wavelengths below 800 nm will have very significant or dominant contribution from the intra-band mechanism, for the large band-gap materials such as quartz.

What is the physical origin of this interplay?

Figure 7: (a-d) lattice site separation-resolved far-field HHG spectra calculated with driving laser wavelengths of $0.6 \mu\text{m}$, $0.8 \mu\text{m}$, $1.0 \mu\text{m}$, and $1.2 \mu\text{m}$, respectively. All spectra are calculated with a driving field beam waist of $\sigma = 100 \mu\text{m}$, a far-field propagation distance of 1 m , a peak driving electric field amplitude of 0.014 a.u. , and are normalized for clarity. The dashed white lines denote the minimum lattice site separation separation for HHG emission at a given harmonic order calculated using Eq. (17) and the vertical dashed red lines denote the harmonic order of the minimum band gap.

Figure 8: The short-time Fourier transforms of the radially integrated far-field HHG spectra shown in Fig. 7 (a-d) calculated with driving laser wavelengths of (a) $0.6 \mu\text{m}$, (b) $0.8 \mu\text{m}$, (c) $1.0 \mu\text{m}$, and (d) $1.2 \mu\text{m}$ showing the above-band-gap HHG emission. All spectra are normalized to their peak above-band-gap emission.

2.4 Physical interpretation of the interplay between inter-band and intra-band mechanisms under different conditions

Intraband currents are initiated by the injection of the charge across the band gap at t_i . These currents are proportional to the number of injected charge carriers, the density $n_c(t_i)$, and their velocity $v(t, t_i)$ dictated by the band dispersion, $j(t, t_i) = n_c(t_i)v(t, t_i)$. In contrast to the intraband currents, the interband contribution is given by the coherence created between the valence and the conduction bands upon electron injection. A simplified expression for the induced polarization is: $P(k; t) = a_c(k, t, t_i)a_v(k, t, t_i)e^{-i \int_{t_i}^t \epsilon(k(\tau))d\tau}$. This expression assumes a fully coherent quantum evolution between injection at t_i and observation at t . Here $a_c(k, t)$, $a_v(k, t)$ are the density *amplitudes*, with $n_c = |a_c|^2$, $n_v = |a_v|^2$.

The main difference between the two processes is simple: in the intraband case the process is dominated by the charge density while the interband process is dominated by coherence. For small injection probabilities per half-cycle, coherence scales with the **square root** of the injection probability per half-cycle, while the intraband competitors scale with the **full** probability. Therein lies the possibility to understand when one or the other dominate. In the case of low frequency driving field and long laser pulses, the dominant mechanism for charge injection into the conduction band is tunneling. In this regime, and for long laser pulses, the injection via optical tunnelling per laser cycle is exponentially low. Therefore, the intraband contributions becomes negligible while the interband contribution dominates the interaction. For few-cycle laser pulses and higher frequencies, the injection rate per laser cycle is necessarily much higher, making the intraband mechanism highly competitive. Since the intraband mechanism also does not suffer from dephasing, it can dominate the resulting far-field emission.

Let us also look at it from another perspective. Harmonic generation, like any parametric process, requires the system to return to its initial state. This applies to both inter-band and intraband contributions. This condition is very clear in the case of interband process (figure 9a). The electron in the conduction band, which tunneled across the band gap with an amplitude a_T , returns with higher energy and recombines across the gap with the hole ($\langle \Psi_v(k, t) | \hat{d} | \Psi_c(k, t) \rangle + \text{c.c.}$). This process is composed of only one tunneling step and scales with a_T . However, the intraband trajectory also must return to the initial state (figure 9b). We can express this condition as a two step process, where the two steps are time-reversed to each other ($\langle \Psi_c(k, t) | \hat{d} | \Psi_c(k, t) \rangle + \text{c.c.}$). The electron tunnels across the band-gap, moves inside the conduction band, and emits harmonic light during this motion. This step, described by $|\Psi_c(k, t)\rangle$, scales with the tunneling amplitude (a_T). Then, the electron *traces its route backwards in time, including the tunneling step*, which is described by $\langle \Psi_c(k, t) |$ and scales with the complex conjugated amplitude a_T^* . In this case the initial and the final states are both associated with the conduction band and therefore they contain the tunnelling amplitude on the both sides: the bra- and the ket- sides (The bra-side is time-reversed to the ket). The result scales with tunnelling probability, $a_T^* a_T = |a_T|^2$.

Thus, as soon as we reduce the frequency of the driving field to bring the interaction into

Figure 9: **Illustration of the interband and intraband mechanisms.** (a), In the interband picture, the electron tunnels with an amplitude a_T , evolves ($|\Psi_c(k, t)\rangle$) and returns to the origin ($\langle \Psi_v(k, t) |$), emitting a photon. (b) In the intraband picture, the electron tunnels with amplitude a_T , evolves ($|\Psi_c(k, t)\rangle$) and returns to the initial state according to the time reversal process ($\langle \Psi_c(k, t) |$), acquiring additional tunneling amplitude a_T^* .

tunnelling or the non-adiabatic tunnelling regime, the interband contribution should dominate the intraband one for high harmonics: the second tunnelling step, back from the conduction to the valence band, is not needed for the interband emission. This explains why the interband contribution usually dominates in the low-frequency driving regime unless the laser pulses are very short (few cycles). Indeed, in the case of a few cycle pulses, the driving field can be very intense without destroying the sample [3, 7]. In fact, ultrashort pulse duration forces one to use high intensities – otherwise the total integrated signal is low. High intensity makes tunnelling ”less painful” and the price for the second – time-reversed – tunnelling event is no longer penalizing. Moreover, since the intraband mechanism does not suffer from dephasing and is also naturally well-collimated in the far-field, it can dominate the resulting emission.

To conclude, the inter-band contribution naturally dominates in the low-frequency regime of optical tunnelling, whether adiabatic or non-adiabatic, because it scales with the tunnelling amplitude per half-cycle and not the tunnelling probability per half-cycle. Detrimental effects of ionization on the macroscopic high harmonic generation strongly limit tunnelling probability per half-cycle in the multi-cycle, long-pulse regime. The intraband contribution can, however, dominate for high harmonics in the nearly single-cycle pulse regime. This regime naturally requires high intensities since there is only a couple of half-cycles that contribute to the signal, and one needs to make sure that enough signal is generated per half-cycle. High intensity lowers the tunnelling penalty. Then the simplicity of the harmonic phase associated with the intraband contribution further enhances its role in the far-field signal.

2.5 Analysis of recent measurements at NRC

Finally, as the referee correctly emphasizes, the NRC group has performed an in-situ measurement in quartz, suggesting that the intraband mechanism dominates the HHG process. We completely agree that this measurement requires a careful comparison with our analysis as well as a proper citation. How can we understand the apparent contradiction between the results of Hammond et al. [8] and our in-situ study?

Hammond et al. studied attosecond pulse generation, driven by a controlled subcycle electric-field transients. They compare the generation mechanism in the gas phase (Xenon) and crystals (quartz). The underlying HHG mechanism is probed via the in-situ scheme. In contrast to the study described in this reply letter, Hammond et al. applied the perturbation by the addition of a weak *fundamental* field in a non-collinear configuration, perturbing the beam's divergence.

As expected, scanning the relative delay between the strong and the perturbative fields in Xenon leads to oscillations of the beam's divergence with a periodicity of ω_0 (the fundamental frequency), where the oscillation phase changes across the spectrum, reflecting the well known atto-chirp. The results in quartz are quite different, and the spectral dependence of the beam divergence is weaker. A deeper insight into the experimental results, presented in the SI, shows that in the case of quartz, scanning the two fields delay leads to modulation of the HHG signal with a periodicity of $2\omega_0$. Such observation cannot be explained by the broken inversion symmetry of the target.

In the following we present a possible interpretation for these results. The authors generate attosecond pulses from x-cut quartz. This crystallographic plane is different from the one in our experiment and other works (z-cut). This plane induces birefringence and phase matched second harmonic generation. Therefore, the propagation of both the fundamental field and the perturbative field in the crystal generates SH as well, which seeds the harmonic's generation process. The oscillation of the beam divergence with $2\omega_0$ periodicity shows that the SH field, generated in the crystal, dominates the perturbative measurement. The flat spectral response encodes the interplay between two nonlinear processes together – the SH generation as well as the HHG. This experimental result is exciting and contains rich information regarding the control of the HHG process via multiple nonlinear processes. However, we believe that the interplay between these processes hinders the ability to isolate the HHG mechanism and identify its underlying dynamics.

In order to avoid the interplay between different nonlinear processes, in our experiment we have generated HHG with z-cut quartz [7,23], while perturbing the interaction with an external SH field. Our scheme enables an independent control of the perturbing field, while allowing a simple and direct understanding of the underlying dynamics.

2.6 Overall conclusions

To summarize our new theoretical and experimental studies, the theoretical description, the numerical simulations, and the new experimental results fully confirm that in our experimental conditions the interband mechanism dominates the HHG process. We thank the referee for raising this concern which initiated this extensive study. We believe that the experimental and theoretical study is not limited to quartz. It provides a general picture, shading light on the interplay between the interband and intraband HHG mechanisms in condense matter systems.

2.7 Changes in the manuscript

Following the referee comment and the findings of our new study, we modified the manuscript and the SI as following:

1. We have written a completely new chapter in the SI (chapter 3). In this chapter we describe in detail the experimental and theoretical studies presented here. We believe that this chapter will provide a broad and deep picture on the fundamental parameters that manipulate the balance between the interband and intraband mechanisms.

2. Following the referee's criticism we have removed from the main text and the SI the direct comparison between the band structure and the harmonic spectrum.

3. We added the following description to the main text (line 140): "In order to obtain a deep understanding of the underlying mechanism that dominates our observations, we combined detailed theoretical analysis with systematic experimental study. Following these studies, we conclude that under our experimental conditions, the interband mechanism dominates the harmonics emission. We note that such an observation is in contrast to previous observation of HHG in quartz [33], performed with shorter wavelength and few-cycle laser pulses. A detailed description of these studies is presented in the SI."

Comment V) The authors argue on the importance of this work with reference to the importance of the Berry phase (the original one). But what they present is not the Berry phase, is something new. Previous works, (for example, Luu et al., Nature communications 2018) have used precisely the same system and it seems that their reconstructed Berry phase matches pretty well the expectations (to the extend a comparison is accurate). There the measured quantity is indeed connected directly to the broadly known Berry phase.

For this work to stand higher than the previous two papers on this topic, this referee would expect that the Berry phases measured would have a more important metrological value, benchmarke/theoretical quantities not captured by the previous works. I feel that this aspect

is missing here and irrespectively of (I) I would feel uncomfortable to support this work without seeing these facts transparently presented.

We completely agree with the referee that the original version of the manuscript did not provide a proper comparison to previous works. More importantly, following your criticism, as well as the review of referee 3, we have realized that we did not provide a clear description of the novelty of our findings. In the following we provide a comprehensive description of our study as well as its comparison to previous studies.

Over the past several years HHG spectroscopy has been applied as a unique probe of electron dynamics in broken inversion symmetry crystals [7, 23–26]. These studies have demonstrated the ability to resolve the *Berry curvature*. In strong contrast, our paper demonstrates, the first (as far as we know) measurement of the *Berry phase*. The ability to reveal the Berry phase, which plays a significant role in condensed matter physics, defines the main novelty of our study.

Indeed, these two quantities are related to each other, however they lead to different phenomena in condensed matter physics [27]. The Berry curvature leads to the appearance of the anomalous velocity and the classical Hall effect. The emergence of the Berry phase leads to quantized electronic phenomena and new quantum phases of matter, such as the Quantum Hall effect and Chern insulator. We totally agree that the original version of the manuscript did not provide a clear introduction of these observables. We have remedied this defect in the new manuscript. Furthermore, below we provide the formalism related to the Berry phase/curvature and describe how they can be resolved experimentally.

When a quantum wavefunction evolves in the parameter space, it accumulates a phase, referred to as the Berry phase (γ_B) [27, 28]. The Berry phase is described by the integration over a vector-valued function, the Berry connection ($\mathcal{A}_n(\mathbf{k}) = i \langle u_{n,\mathbf{k}}(x) | \nabla_{\mathbf{k}} | u_{n,\mathbf{k}}(x) \rangle$). The Berry connection, in every realization, is not a gauge invariant quantity and therefore it is not an observable. Unlike the Berry connection, in the particular case of a closed loop, the Berry phase becomes gauge invariant and can be measured. In this case, the Berry phase represents a geometrical phase; a **global** quantity that depends on the contour of the evolution only, $\gamma_B(C)$. In analogy to gauge field theories, one can define a third quantity, the Berry curvature, $\mathbf{\Omega}(k) = \nabla_{\mathbf{k}} \times \mathcal{A}$, which represents the flux inside the loop (figure 10). In contrast to the Berry phase, the Berry curvature is a **local** (rather than global) gauge invariant quantity, defined at every point in the parameter space, and can be measured even without a closed loop contour. The three quantities are related according to:

$$\gamma_B(C) = \int_C d\mathbf{k} \mathcal{A}(\mathbf{k}) = \int_S d^2\mathbf{k} \mathbf{\Omega}(\mathbf{k}) \quad (18)$$

While, both the Berry phase and the Berry curvature can be measured, there are important differences between them:

Figure 10: **Comparison between Berry phase and Berry curvature.** While the Berry curvature is defined at every point (square) in the parameter space ($\Omega(k)$, blue and pink squares), the Berry phase, $\gamma_B(C)$, depends on the closed path (black contour).

1. The Berry curvature is a local property, defined at every point in k space (figure 10). Importantly, the Berry curvature does not depend on the evolution of the wavepacket. In contrast, the Berry phase depends on the contour of the evolution of the wavefunction. Therefore, in contrast to the Berry curvature, it is not a static property of the medium but a dynamical one that varies with the driving field. Therefore, one cannot define general characteristics of the Berry phase in quartz.

2. The Berry curvature can be directly measured via the transverse polarization it induces in light matter interactions, or via transversal resistivity (Hall resistivity) in transport measurements. Such transverse polarization can be resolved in any system, as long as the inversion symmetry or time reversal symmetry is broken [23].

However, resolving the Berry phase requires an interferometric scheme. Performing an interferometric measurement in condensed matter systems is extremely challenging, due to dephasing effects abundant in many-body systems. Electron-phonon, electron-electron interaction and scattering from impurities can lead to loss of the phase information. One of the most important breakthroughs of this field was accomplished by achieving very clean samples of GaAs-AlGaAs heterostructures, establishing 2D electron gas system at extremely low temperature (mK scaling) [29, 30]. The coherence is achieved by fabricating high quality devices with high mobility. Only recently, interferometry experiments in intrinsic electronic systems were performed in very clean graphene samples, at 60 mK [31]).

In contrast, in this work we resolve for the first time the Berry phase, in a trivial insulator, under room temperature conditions.

In our paper we achieve coherence by performing the interferometric measurement on an attosecond time scale, before any scattering or dephasing mechanism evolves. We believe that

the ability to resolve the quantum nature of the wavefunction on extremely short time scales is one of the most important advantages of attosecond science in condensed matter systems.

Changes made: Following the referee comment, we have added a new section to the SI (chapter 5.5), where we provide a proper introduction and discuss in details the fundamental difference between these two observables. In addition, we added a clarification in the manuscript (line 199): "While the Berry phase and the Berry curvature are strongly related, their physical properties are inherently distinct, leading to different observations (see SI)". Finally, we compare our work with previous work as well as provide a proper citation (line 99): "The short time scale of the mechanism provides the key advantage of our scheme. Schmid et al. Ref [16] revealed the Berry curvature of topological insulators via HHG driven by THz field, having a fundamental period of 40fs. In their study, topology enables to overcome dephasing and scattering mechanisms, revealing the geometrical properties of the system. Our measurement, performed on an attosecond time scale, allows the probing of the Berry phase in trivial insulator.

Comment VI: More technically. The authors assume that the "excitation" matrix element is independent of the polarization state of the strong pulse. In this sense the polarized field is simply decomposed to two perpendicular fields that guide the motion of the excited carrier(s) only. Is there experimental evidence that this is the case? Alternatively, is it supported by rigorous calculations?

We thank the referee for raising this important question. Indeed, the variation of the excitation matrix with the driving field polarization cannot be neglected. Our model, which considers the "generalized Berry phase", combines the common Berry phase notation, the integration over the Berry connection, with the excitation matrix phase. A full description is presented in chapter 5.1 in the SI, here we present the main points.

In a broken inversion system, the dipole coupling is a complex quantity, composed by real and imaginary components. Adding the dipole phases to the action, defining the *generalized action*, provides a gauge invariant quantity, as presented in the main text (Equation 1):

$$J_{inter}^{(i)}(\omega) = i \int_{BZ} d^3K \int_{-\infty}^t dt' \int_{-\infty}^{\infty} dt F^{(j)}(t') \cdot |d^{(j)}(\mathbf{K} + \mathbf{A}(t'))| |d^{*(i)}(\mathbf{K} + \mathbf{A}(t))| e^{-iS^{(i)}[\mathbf{K}, t', t] + i\omega t} + c.c$$

$$S^{(i)}[\mathbf{K}, t', t] \equiv \int_{t'}^t [\varepsilon_g(\mathbf{K} + \mathbf{A}(\tau)) + \mathbf{F}(\tau) \cdot \mathcal{A}_g(\mathbf{K} + \mathbf{A}(\tau))] d\tau + \phi_d^{(j)}(\mathbf{K} + \mathbf{A}(t')) - \phi_d^{(i)}(\mathbf{K} + \mathbf{A}(t))$$
(19)

Here $\mathbf{d}(\mathbf{k}) = i \langle u_{v,\mathbf{k}}(x) | \nabla_{\mathbf{k}} | u_{c,\mathbf{k}}(x) \rangle$ is the dipole coupling between the valence and the conduction bands, $u_{v/c,\mathbf{k}}(x)$ is the periodic part of the Bloch functions ($\psi_{n,\mathbf{k}} = e^{ik \cdot r} u_{n,\mathbf{k}}(r)$) and $\varepsilon_g = \varepsilon_c - \varepsilon_v$ is the two bands energy difference. $\mathcal{A}_g = \mathcal{A}_c - \mathcal{A}_v$ is their relative Berry connection, defined as $\mathcal{A}_{\mathbf{m}}(\mathbf{k}) = i \langle u_{m,\mathbf{k}}(x) | \nabla_{\mathbf{k}} | u_{m,\mathbf{k}}(x) \rangle$, in which $m = v, c$. $\mathbf{F}(t)$ and $\mathbf{A}(t)$ are the laser field and its vector potential and $\mathbf{K}(\mathbf{t}) = \mathbf{k} - \mathbf{A}(t)$ is the canonical crystal quasi-momentum.

ϕ_d is the dipole coupling phase and $j, i = x, y$.

We separate the action into its trivial component, which controls the interaction in inversion symmetric crystal and to the gauge invariant Berry phase:

$$\begin{aligned}
S^{(i)}[\mathbf{K}, t', t] &\equiv S_{inv} + \gamma_B^{(i)} \\
S_{inv}[\mathbf{K}, t', t] &\equiv \int_{t'}^t \varepsilon_g(\mathbf{K} + \mathbf{A}(\tau)) d\tau \\
\gamma_B^{(i)}[\mathbf{K}, t', t] &\equiv \int_{t'}^t \mathbf{F}(\tau) \cdot \mathcal{A}_g(\mathbf{K} + \mathbf{A}(\tau)) d\tau + \phi_d^{(j)}(\mathbf{K} + \mathbf{A}(t')) - \phi_d^{(j)}(\mathbf{K} + \mathbf{A}(t))
\end{aligned} \tag{20}$$

Due to time reversal symmetry, S_{inv} is symmetric at positive and negative crystal momentum, $S_{inv}(-\mathbf{k}) = S_{inv}(\mathbf{k})$, while the Berry phase switches its sign, $\gamma_B(-\mathbf{k}) = -\gamma_B(\mathbf{k})$. Finally, in order to simplify the time dependence of this expression we rewrite the Berry phase as:

$$\begin{aligned}
\gamma_B^{(i)}[\mathbf{K}, t', t] &\equiv \tilde{\gamma}_B + \Delta\phi_d^{(i)} \\
\tilde{\gamma}_B &\equiv \int_{t'}^t \mathbf{F}(\tau) \cdot [\mathcal{A}_g(\mathbf{K} + \mathbf{A}(\tau)) + \nabla_K \phi_d^{(j)}(\mathbf{K} + \mathbf{A}(\tau))] d\tau \\
\Delta\phi_d^{(i)} &\equiv \phi_d^{(j)}(\mathbf{K} + \mathbf{A}(t)) - \phi_d^{(j)}(\mathbf{K} + \mathbf{A}(t'))
\end{aligned} \tag{21}$$

We define $\tilde{\gamma}_B$ as a *generalized Berry phase*, which is a gauge invariant quantity that **contains the excitation dipole phase**. The contribution of the excitation matrix is crucial for the physical meaning of this quantity. Its contribution, $\nabla_K \phi_d^{(j)}(\mathbf{K} + \mathbf{A}(\tau))$ depends on the driving field polarization in the same manner as the other term that contribute to the generalized Berry phase. In our experiment we resolve the generalized Berry phase that includes the variation of the excitation matrix with the driving field polarization.

Changes made: Following your comment we have re-written chapter 5.1 in the SI and clarified the contribution of the excitation matrix to the generalized Berry phase.

3 referee 3

The data are clear, the analysis is sound, and the results are exciting. However, as spectacular as the results may seem assuming a first observation, there is evident overlap with existing works. Schmid et al., cited (as [40]) by the authors, have already presented direct measurement of the Berry phase in a topological insulator at lower frequencies from mid-IR to VIS/UV. It is unclear why this reference is somewhat neglected by placing it at the very end of the main text rather than discussing the similarity with the present data. To be clear: the differences are interband as well as intraband Berry phases on the one hand and intraband-only Berry phases on the other, while the similarities include ballistic lightwave acceleration, high-harmonics generation, HHG interference, Berry curvature in general, the Berry curvature shown in Fig. 4c and [40] in particular, subcycle dynamics, polarization effects in HHG, Bloch's acceleration theorem, solid-state all-coherent polarization dynamics, and more. In this light, the authors would have to argue why their work is not an extension of previous works, that merely adds, technicalities aside, the interband Berry connection (although that result would still be quite interesting). While I like the results, I must ask for their placement to be revised. Under the current circumstances, I would not support publication in Nature owing to the limited degree of novelty as described above. Moreover, I list a few more issues below which the authors can certainly address.

We thank the referee for his very positive evaluation of our experimental and theoretical work. The main concern of the referee is the novelty of our work compared to the (truly exceptional) work of Schmid et al. [32], as well as Refs. [23, 24, 26]. We would like to argue that our work reports a fundamental advance compared to previous works, as our study focuses on the Berry phase, while Schmid et al. have focused on resolving the Berry curvature. We attribute this misunderstanding to the drawbacks of the first version of our manuscript, and we have made significant modifications to better place our work in the context of the previous studies [7, 23–26], and especially of the recent discovery by Schmid et al [32].

Specifically, Schmid et al. [32] observed high harmonic emission using THz driving field, generated from the surface as well as from the bulk 3D topological insulator (3D TI). This outstanding work shows the ability to isolate and control the contribution of the surface states to the harmonic emission. An evidence of their contribution is the generation of even harmonics, due to the appearance of large Berry curvature at the surface.

In the following we discuss the fundamental differences between the two studies:

- Schmid et al connected their measurements of HHG in a topological insulator to the geometrical properties of the system focusing on the (intra-band) Berry curvature, but not on the intrinsic Berry phase of system. In strong contrast, our work focuses on the Berry phase in a trivial insulator and introduces a new type of the Berry phase, which corresponds to a closed loop both in continuous (intraband) and discrete (interband) variables.

Figure 11: **Intrinsic Berry phase and light driven Berry phase.** (a) The intrinsic Berry phase appears at the surface states of 3D topological insulator, accumulated by integrating over a loop around the Dirac point (this picture is taken from ref [33]). (b) The light driven interband Berry phase, introduced in our work, accumulated during the evolution in a closed path between the bands.

- While these two quantities are related to each other, $\gamma_B(C) = \int_C d\mathbf{k} \mathcal{A}(\mathbf{k}) = \int_S d^2\mathbf{k} \Omega(\mathbf{k})$, they are fundamentally distinct, leading to different observations in condensed matter physics: The Berry curvature, which can also be found in trivial insulators [24], is the origin of the anomalous velocity and the valley degree of freedom [26]. The Berry phase leads to **quantized** electronic phenomena and topological phases, e.g. the Quantum Hall effect and the Chern insulator [27]. For example, the value of the intrinsic Berry phase in the system studied in the work of Schmid et al. must be π , while the Berry curvature that was experimentally resolved presenting different values. For a more detailed discussion see our answer to referee 2.
- The Berry curvature can be directly measured via the transverse polarization it induces in light matter interactions or via transverse resistivity (Hall resistivity) in transport measurements. Resolving the Berry phase requires an interferometric scheme.
- Performing an interferometric measurement in condensed matter systems is extremely challenging, due to multiple dephasing channels found in many-body systems. Only recently, interferometry experiments in intrinsic electronic systems were performed in very clean graphene samples, at 60 mK [31]). To the best of our knowledge, in this work we resolve, for the first time, a quantum phase in a trivial insulator under room temperature conditions.
- The key to overcoming decoherence effects in our approach is the use of a different time scale compared to Schmid et al. Ref [32] focuses on HHG driven by THz field, having a fundamental period of $40fs$. On this time scale, dephasing and scattering impose significant limitations. Quoting from ref [32]: "efficient scattering and dephasing destroys

the electronic quantum memory on extremely short timescales setting narrow limits to strong-field quantum control”. Topology enables the authors to overcome this challenge: ”Non-trivial topology promises a paradigm change”. Our study overcomes this challenge with a completely different approach: We perform the measurement on a time scale one order of magnitude faster, such that scattering and dephasing mechanisms play a minimal role.

We believe that these two studies do not overlap, but rather complement each other, reflecting various approaches and aspects in this rapidly evolving field of research.

Changes made: Following the referee comment, we have added a new section to the SI (chapter 5.5), where we provide a proper introduction and discuss in details the fundamental difference between these two observables. In addition, we added a clarification in the manuscript (line 199): ”While the Berry phase and the Berry curvature are strongly related, their physical properties are inherently distinct, leading to different observations (see SI)”. Finally, we compare our work with previous work as well as provide a proper citation (line 99): ”The short time scale of the mechanism provides the key advantage of our scheme. Schmid et al. Ref [16] revealed the Berry curvature of topological insulators via HHG driven by THz field, having a fundamental period of 40fs. In their study, topology enables to overcome dephasing and scattering mechanisms, revealing the geometrical properties of the system. Our measurement, performed on an attosecond time scale, allows the probing of the Berry phase in trivial insulator”.

Comment: Validity of equation 1 is not straight-forward or intuitive to see. Some terms connect valence and conduction bands, others are intraband, without an easily recognizable pattern. The authors should discuss the structure more in detail.

Indeed, equation 1 is at the heart of our study, capturing the physical picture of the *Interband Berry phase*, which is introduced here for the first time. A detailed mathematical description is presented in the SI, chapter 5.4, here we present the key steps discussing its structure and physics. The left side of the equation 1 reads:

$$\lim_{N \rightarrow \infty} \langle \mathbf{u}_{v, \mathbf{k}_1} | \mathbf{u}_{c, \mathbf{k}_2} \rangle \langle u_{c, \mathbf{k}_2} | u_{c, \mathbf{k}_3} \rangle \cdots \langle \mathbf{u}_{c, \mathbf{k}_{N-1}} | \mathbf{u}_{v, \mathbf{k}_N} \rangle \cdots \langle u_{v, \mathbf{k}_2} | u_{v, \mathbf{k}_1} \rangle \quad (22)$$

where $|u_{n, \mathbf{k}}\rangle$ is the periodic part of the Bloch function ($n = v, c$ for the valence and conduction bands). Eq.(1) describes the evolution of a wavefunction in a closed loop, generalizing the loop to include not only continuous, but both continuous and discrete variables. The wavefunction starts at $\langle u_{v, \mathbf{k}_1} |$ (the first ”bra”) and comes back to the same state $|u_{v, \mathbf{k}_1}\rangle$ (the last ”ket”). The *interband Berry phase* is associated with this closed loop in the energy – momentum space, with the evolution of the wavefunction including both propagation within the bands and transitions between the bands.

We can separate the loop into four parts, as shown in the Figure 12. First, the electron

Figure 12: **Illustration of equation 1 in the main text.** (a) The evolution of the wavepacket, along and between the bands, notated by wavefunction $|u_{n,k}\rangle$ ($n = v, c$). (b) A points loop presentation for (a), where the purple line represents continuous evolution while the green and the blue lines represent the discrete evolution. (c) The associated phase for every step along the loop.

tunnels from the valence band to the conduction band (interband transition), associated with the term $\langle u_{v,k_1} | u_{c,k_2} \rangle$. Second, the electron propagates within the conduction band (intraband evolution), changing its momentum from k_2 to k_3 , etc, until k_{N-1} . This evolution is described by $\langle u_{c,k_2} | u_{c,k_3} \rangle \cdot \langle u_{c,k_3} | u_{c,k_4} \rangle \cdot \langle u_{c,k_4} | u_{c,k_5} \rangle \cdots \langle u_{c,k_{N-2}} | u_{c,k_{N-1}} \rangle$. Third, the electron recombines with the hole via the interband transition, associated with $\langle u_{c,k_{N-1}} | u_{v,k_N} \rangle$. The time-reversed evolution of the hole in the bra closes the loop: the hole evolves within the valence band from $\langle u_{v,k_N}$ back to $|u_{v,k_1}\rangle$. The Figure 12 describes this evolution pictorially.

The right hand side of equation 1 shows how this evolution is recorded by the accumulated Berry phase. The phase of the term

$$e^{i \int_{t'}^t \varepsilon_g(\mathbf{k}(\tau)) d\tau + i \gamma_{B,int}} \quad (23)$$

contains the dynamical phase and the Berry phase, the latter given by

$$\gamma_{B,int} \equiv \int_{t'}^t \mathbf{F}(\tau) \cdot (\mathcal{A}_g(\mathbf{k}(\tau)) + \nabla_{\mathbf{k}} \phi_d(\mathbf{k}(\tau))) d\tau$$

Here $\varepsilon_g = \varepsilon_c - \varepsilon_v$ is the band gap, $\mathcal{A}_g = \mathcal{A}_c - \mathcal{A}_v$ is the electron-hole relative Berry connection ($\mathcal{A}_n(\mathbf{k}) = i \langle u_{n,\mathbf{k}}(x) | \nabla_{\mathbf{k}} | u_{n,\mathbf{k}}(x) \rangle$), and $\phi_d(\mathbf{k})$ is the phase of the interband dipole coupling: $\phi_d(\mathbf{k}) = \arg(i \langle u_{v,\mathbf{k}}(x) | \nabla_{\mathbf{k}} | u_{c,\mathbf{k}}(x) \rangle)$. The crystal momentum $\mathbf{k}(\tau) = \mathbf{k} - \mathbf{A}(t) + \mathbf{A}(\tau)$ is controlled by the laser field, where $\mathbf{F}(t)$ and $\mathbf{A}(t)$ are the laser electric field and the vector potential. The instants t and t' define the transition times between the bands (the ‘‘ionization’’, or ‘injection’, and the recombination times).

We can transform the integral over time to the integral over momenta, taking into account

that $d\mathbf{k}(\tau) = -\mathbf{F}(\tau)d\tau$:

$$\gamma_{B,int} = - \int_{\mathbf{k}_i}^{\mathbf{k}_f} [\mathcal{A}_c(\mathbf{k}) - \mathcal{A}_v(\mathbf{k})]d\mathbf{k} + \phi_d(\mathbf{k}_f) - \phi_d(\mathbf{k}_i). \quad (24)$$

This phase indeed captures the four steps of the energy- momentum loop: the phase associated with the injection (phase of the dipole at the ionization momentum, $\phi_d(k_i)$), the phase associated with the intraband evolution of the electron ($\int_{\mathbf{k}_i}^{\mathbf{k}_f} \mathcal{A}_c(\mathbf{k})d\mathbf{k}$), the phase of the recombination dipole ($\phi_d(k_f)$), and finally the intraband evolution of hole ($-\int_{\mathbf{k}_i}^{\mathbf{k}_f} \mathcal{A}_v(\mathbf{k})d\mathbf{k}$).

Changes made: We hope that the detailed description clarifies the origin of equation 1. Following your comments, we have made the following changes to the manuscript: We have added a detailed mathematical description and the derivation of eq. 1 to the SI, chapter 5.4. In addition, we have added to this chapter the presented figure, giving the pictorial description, together with the associated physical interpretation.

Comment: The presentation of the data in Fig. 2 is not clear. The dashed lines suggest magnifications of areas of the spectra of panel c, but they actually show two cuts at the discrete harmonics. Moreover, panel e shows curves in the background which seem like the log of the intensity. No corresponding description is given. Then, the phase is shown by means of the real and imaginary part of $\Delta\gamma_B$, but for which harmonic? It would make sense to analyze the even harmonics, which also seems to be the case. Also, since the distinction of odd and even harmonics is crucial, the spectra in b should be labelled accordingly (rather than just energy in eV, which almost coincides with the harmonic number, somewhat by accident, since 1200 nm 1 eV, the driving field energy).

Somewhat disappointingly, the color plot of Fig. 3c does not have a scale bar. It is thus, strictly speaking, worthless, as there could be any mapping of color and intensity.

One of the early investigations of this field which established the mechanism of coherent interband excitation and intraband transport is missing (Schubert et al., Nature Photonics 8, 119 (2014)).

We completely agree with your criticism, indeed the original presentation of these figures was unclear. Following your comments we have modified figures 2 and 3 and included the following changes:

- In figure 2, we changed the presentation of the data to keep the same logarithmic scale through all the sub-figures. We added colorbar to figure 2b and 2c, as mention in the caption as well.
- In figure 2d, we plotted the extracted signal of individual harmonics, at logarithmic scale (I_{odd} or I_{even}) and added their number as well. We added left and right y axes and a

corresponding description in the caption.

- In figure 2e, we kept the same plots that are presented in figure 2d, in the background (to avoid any confusion), and we added right y axis for the imaginary Berry phase. We also changed the colors of the plots for better visibility.
- We have added the colorbar to figure 3.
- We have added the reference to Schubert et al.

We thank you for this important input.

References

- [1] Lun Yue and Mette B Gaarde. Imperfect recollisions in high-harmonic generation in solids. *Physical Review Letters*, 124(15):153204, 2020.
- [2] KS Budil, P Salières, Anne L’Huillier, T Ditmire, and MD Perry. Influence of ellipticity on harmonic generation. *Physical Review A*, 48(5):R3437, 1993.
- [3] Manish Garg, Minjie Zhan, Tran Trung Luu, H Lakhotia, Till Klostermann, Alexander Guggenmos, and Eleftherios Goulielmakis. Multi-petahertz electronic metrology. *Nature*, 538(7625):359, 2016.
- [4] Shambhu Ghimire, Anthony D DiChiara, Emily Sistrunk, Pierre Agostini, Louis F DiMauro, and David A Reis. Observation of high-order harmonic generation in a bulk crystal. *Nature physics*, 7(2):138, 2011.
- [5] Tran Trung Luu, M Garg, S Yu Kruchinin, Antoine Moulet, M Th Hassan, and Eleftherios Goulielmakis. Extreme ultraviolet high-harmonic spectroscopy of solids. *Nature*, 521(7553):498, 2015.
- [6] G Vampa, TJ Hammond, N Thiré, BE Schmidt, F Légaré, CR McDonald, T Brabec, and PB Corkum. Linking high harmonics from gases and solids. *Nature*, 522(7557):462, 2015.
- [7] Manish Garg, Hee-Yong Kim, and Eleftherios Goulielmakis. Ultimate waveform reproducibility of extreme-ultraviolet pulses by high-harmonic generation in quartz. *Nature Photonics*, 12(5):291–296, 2018.
- [8] TJ Hammond, DM Villeneuve, and PB Corkum. Producing and controlling half-cycle near-infrared electric-field transients. *Optica*, 4(7):826–830, 2017.

- [9] Yong Sing You, Yanchun Yin, Yi Wu, Andrew Chew, Xiaoming Ren, Fengjiang Zhuang, Shima Gholam-Mirzaei, Michael Chini, Zenghu Chang, and Shambhu Ghimire. High-harmonic generation in amorphous solids. *Nature communications*, 8(1):724, 2017.
- [10] Nicola Marzari, Arash A Mostofi, Jonathan R Yates, Ivo Souza, and David Vanderbilt. Maximally localized wannier functions: Theory and applications. *Reviews of Modern Physics*, 84(4):1419, 2012.
- [11] REF Silva, F Martín, and M Ivanov. High harmonic generation in crystals using maximally localized wannier functions. *Physical Review B*, 100(19):195201, 2019.
- [12] EI Blount. Formalisms of band theory. In *Solid state physics*, volume 13, pages 305–373. Elsevier, 1962.
- [13] N Dudovich, Olga Smirnova, J Levesque, Yu Mairesse, M Yu Ivanov, DM Villeneuve, and Paul B Corkum. Measuring and controlling the birth of attosecond xuv pulses. *Nature physics*, 2(11):781, 2006.
- [14] O Pedatzur, G Orenstein, V Serbinenko, H Soifer, BD Bruner, AJ Uzan, DS Brambila, AG Harvey, L Torlina, F Morales, O Smirnova, and N Dudovich. Attosecond tunnelling interferometry. *Nature Physics*, 11(10):815, 2015.
- [15] Ayelet J Uzan, Hadas Soifer, Oren Pedatzur, Alex Clergerie, Sylvain Larroque, Barry D Bruner, Bernard Pons, Misha Ivanov, Olga Smirnova, and Nirit Dudovich. Spatial molecular interferometry via multidimensional high-harmonic spectroscopy. *Nature Photonics*, pages 1–7, 2020.
- [16] D Shafir, Y Mairesse, DM Villeneuve, PB Corkum, and N Dudovich. Atomic wavefunctions probed through strong-field light-matter interaction. *Nature Physics*, 5(6):412, 2009.
- [17] Ayelet Julie Uzan, Gal Orenstein, Álvaro Jiménez-Galán, Chris McDonald, Rui EF Silva, Barry D Bruner, Nikolai D Klimkin, Valerie Blanchet, Talya Arusi-Parpar, Michael Krüger, et al. Attosecond spectral singularities in solid-state high-harmonic generation. *Nature Photonics*, 14(3):183–187, 2020.
- [18] Giulio Vampa, Jian Lu, Yong Sing You, Denitsa R Baykusheva, Mengxi Wu, Hanzhe Liu, Kenneth J Schafer, Mette B Gaarde, David A Reis, and Shambhu Ghimire. Attosecond synchronization of extreme ultraviolet high harmonics from crystals. *Journal of Physics B: Atomic, Molecular and Optical Physics*, 53(14):144003, 2020.
- [19] Yong Sing You, Mengxi Wu, Yanchun Yin, Andrew Chew, Xiaoming Ren, Shima Gholam-Mirzaei, Dana A Browne, Michael Chini, Zenghu Chang, Kenneth J Schafer, et al. Laser waveform control of extreme ultraviolet high harmonics from solids. *Optics letters*, 42(9):1816–1819, 2017.

- [20] Ayelet J Uzan-Narovlansky, Álvaro Jiménez-Galán, Gal Orenstein, Rui EF Silva, Talya Arusi-Parpar, Sergei Shames, Barry D Bruner, Binghai Yan, Olga Smirnova, Misha Ivanov, et al. Observation of light-driven band structure via multiband high-harmonic spectroscopy. *Nature Photonics*, 16(6):428–432, 2022.
- [21] Mengxi Wu, Dana A. Browne, Kenneth J. Schafer, and Mette B. Gaarde. Multilevel perspective on high-order harmonic generation in solids. *Phys. Rev. A*, 94:063403, Dec 2016.
- [22] F. Catoire, H. Bachau, Z. Wang, C. Blaga, P. Agostini, and L. F. DiMauro. Wannier representation of intraband high-order harmonic generation. *Phys. Rev. Lett.*, 121:143902, Oct 2018.
- [23] Tran Trung Luu and Hans Jakob Wörner. Measurement of the berry curvature of solids using high-harmonic spectroscopy. *Nature communications*, 9(1):1–6, 2018.
- [24] Hanzhe Liu, Yilei Li, Yong Sing You, Shambhu Ghimire, Tony F Heinz, and David A Reis. High-harmonic generation from an atomically thin semiconductor. *Nature Physics*, 13(3):262, 2017.
- [25] Matthias Hohenleutner, Fabian Langer, Olaf Schubert, Matthias Knorr, U Huttner, SW Koch, M Kira, and Rupert Huber. Real-time observation of interfering crystal electrons in high-harmonic generation. *Nature*, 523(7562):572, 2015.
- [26] Fabian Langer, Christoph P Schmid, Stefan Schlauderer, Martin Gmitra, Jaroslav Fabian, Philipp Nagler, Christian Schüller, Tobias Korn, PG Hawkins, JT Steiner, et al. Lightwave valleytronics in a monolayer of tungsten diselenide. *Nature*, 557(7703):76–80, 2018.
- [27] Di Xiao, Ming-Che Chang, and Qian Niu. Berry phase effects on electronic properties. *Reviews of modern physics*, 82(3):1959, 2010.
- [28] Michael Victor Berry. Quantal phase factors accompanying adiabatic changes. *Proceedings of the Royal Society of London. A. Mathematical and Physical Sciences*, 392(1802):45–57, 1984.
- [29] Amnon Yacoby, Moty Heiblum, Diana Mahalu, and Hadas Shtrikman. Coherence and phase sensitive measurements in a quantum dot. *Physical review letters*, 74(20):4047, 1995.
- [30] Yang Ji, Yunchul Chung, D Sprinzak, Moty Heiblum, Diana Mahalu, and Hadas Shtrikman. An electronic mach–zehnder interferometer. *Nature*, 422(6930):415–418, 2003.
- [31] Yuval Ronen, Thomas Werkmeister, Danial Haie Najafabadi, Andrew T Pierce, Laurel E Anderson, Young Jae Shin, Si Young Lee, Young Hee Lee, Bobae Johnson, Kenji Watanabe, et al. Aharonov–bohm effect in graphene-based fabry–pérot quantum hall interferometers. *Nature nanotechnology*, 16(5):563–569, 2021.

- [32] Christoph P Schmid, Leonard Weigl, P Grössing, Vanessa Junk, Cosimo Gorini, Stefan Schlauderer, Suguru Ito, Manuel Meierhofer, Niklas Hofmann, Dmitry Afanasiev, et al. Tunable non-integer high-harmonic generation in a topological insulator. *Nature*, 593(7859):385–390, 2021.
- [33] Yoshinori Tokura, Kenji Yasuda, and Atsushi Tsukazaki. Magnetic topological insulators. *Nature Reviews Physics*, 1(2):126–143, 2019.

Reviewer Reports on the First Revision:

Referees' comments:

Referee #1 (Remarks to the Author):

The presentation of the inter-Berry phase is more clear now, especially the attosecond interferometric concept to extract the inter-Berry phase sounds convincing to me. I recommend the paper for publication and recommend the authors consider citing:

<https://link.aps.org/doi/10.1103/PhysRevB.102.134115> in the introduction or in the section where they use the CD.

The paper entitled "Observation of interband Berry phase in laser driven Crystal" by Ayelet J. U.- N. et al., present a novel method to extract berry phase differences via attosecond interferometry. "The manuscript shows a conceptually new formalism of the Berry phase, accumulated in both discrete and continuous space. HHG spectroscopy allows us to realize Berry phase interferometry and probe the coherent properties of electron-hole wavefunction on a sub cycle time scale. The authors experimentally demonstrate this scheme and resolve, for the first time, the generalized Berry phase across a large energy range. Extension of the approach to a two color field enables sensitive probing of the Berry curvature."

These experimental and theoretical results are impressive. I think the paper is well presented and encouraging. However, I have a few concerns before recommend it for publication.

- Usually the Berry phase is defined in a closed path, however the interband Berry phase does not seem to follow a closed loop linear integral. Can the author comments on it, please?
- I don't understand Eq. 2 very well, could the authors provide a more detailed explanation where is this Eq. 2 coming from and its relation with the Circular Dichroism?
- At the beginning once we read the paper, we are thinking in a topological material, but alphaquartz is not, the author should mention it clearly.

After this two minor points are fixed, I will recommend publication at Nature.

Referee #2 (Remarks to the Author):

I certainly consider the effort taken by the authors to investigate the basic assumptions of their study both theoretically and experimentally a very essential step forward. Of primary importance I consider the experimental study as theory still needs to further develop to capture the experiments at a satisfactory level.

The authors have conducted a series of omega-2 omega interferometric experiments to demonstrate the presence of atto-chirp on high harmonics. The technique was introduced by some of the authors in atomic systems some 20 years back.

I have difficulties to understand how the original 2D color plots (Supplementary Fig.8, bottom) are translated into the data of (blue lines of Fig. 9). Looking at the false color data it is, for me at least, nearly impossible to discern a chirp. For instance, in the original work from Dudovich et al., 2004 the atto-chirp is rather obvious on the 2D data leaving no room for questions.

Here this is not the case. The argument is then made on further analysis of the data which, at least in my eyes, is unclear regarding accuracy and errors. For such a small chirp demonstrated (blue lines on supplementary Fig. 9) it is of utmost importance that the analysis is carefully done and presented, as there is no doubt that the experiments will be repeated by other groups.

Do the authors integrate over the harmonic signals versus delay to produce lines like the blue line in Fig.8 (bottom)? They probably do. And then, how is the phase derived? A peak-to-peak comparison is not possible as the data are rather noisy and irregular. Do they fit a sinusoidal curve to derive the phase and if so, what is the error of the phase. Note that the claimed phase variation is less than 0.5 cycles!

Moreover, I see that the authors are somewhat selective on the harmonics that they analyze to make their case both on experiment as well as on theory. The even harmonics in their spectrum span from the 12th to the 21st harmonic but the authors analysis only for a few of them around the middle range. Why is this the case? In fact, the higher harmonics have even stronger signals, why would one disregard them so easily. Of course, they authors could argue, "yes but only the ones shown are relevant for the band we study". Yet this is not true in nonlinear optics. Transitions to higher bands do also create harmonics at the energy range of the "probed bands" how do we know if these are so weak, and they do not mess up conclusions?

I suggest the authors to revisit the analysis for all three data sets acquired at three intensities. All the data shall be shown on false color plot as well as evaluated lines and their fittings for all harmonics. The final phases shall be presented with an errorbar so that readers will be allowed to evaluate better the accuracy of the claims. If the claims made here, survive this analysis this would be ideal.

The comparison between theory and experiment in Fig.9 shall be done in a transparent way. For the same energy range. Clearly this not the case now and as a result the argument is not supported.

The authors argue that few-cycle pulses need higher intensity and therefore these experiments will miss the interband picture. The argue that it is all interband but at high intensities the chirp reduces and apparently this makes it to look like an intraband. I do not agree with this argument.

In Fig.9 C the blue line(if we believe that the error bar is the size of the point) we have an irregular curve, this is not a straight line that would be identified as interband or intraband. To me it looks more like noise.

Last the authors argue that they used lower intensity than other studies. I would argue that they use the highest intensity in comparison to all previous time resolved studies.

To my regret despite the mounting evidence that the distortions of the chirp data is dramatic when

the reflective and transmission geometry are used the authors have still opted for a transmission study. Whereas their samples are some ~ 20 μm thick, it is difficult for me to assess how important this is, given that the demonstrated degree of chirp with which the authors are supporting their argument is negligibly small. What would the authors comment on this serious issue?

I believe that the authors recognize here a range of weaknesses that need to be addressed.

Referee #3 (Remarks to the Author):

After almost one year of time, the Dudovich group and collaboration partners have submitted a revised version of their manuscript on the interband Berry phase probed by attosecond interferometry.

In their response letter they state that they have dedicated extensive efforts to addressing the concerns of the referees. For my part, their responses have been fully satisfactory and I now recommend publication of the beautiful manuscript in Nature. The results are extraordinary.

As a final remark, I just noticed that one of the references the authors claimed to have added is not to be found in the manuscript nor in the supplementary information.

Author Rebuttals to First Revision:

1 Referee 1

The presentation of the inter-Berry phase is more clear now, especially the attosecond interferometric concept to extract the inter-Berry phase sounds convincing to me. I recommend the paper for publication and recommend the authors consider citing:
<https://link.aps.org/doi/10.1103/PhysRevB.102.134115> in the introduction or in the section where they use the CD.”

We thank the referee for his careful reading, valuable comments and support in the acceptance of our paper to Nature.

In addition, we completely agree with the referee, indeed this reference is very important and relevant to our study. Following the referee’s comment we added this reference to our manuscript.

2 Referee 2

Comment 1:

I certainly consider the effort taken by the authors to investigate the basic assumptions of their study both theoretically and experimentally a very essential step forward. Of primary importance I consider the experimental study as theory still needs to further develop to capture the experiments at a satisfactory level.

The authors have conducted a series of omega-2 omega interferometric experiments to demonstrate the presence of atto-chirp on high harmonics. The technique was introduced by some of the authors in atomic systems some 20 years back. I have difficulties to understand how the original 2D color plots (Supplementary Fig.8, bottom) are translated into the data of (blue lines of Fig. 9). Looking at the false color data it is, for me at least, nearly impossible to discern a chirp. For instance, in the original work from Dudovich et al., 2004 the atto-chirp is rather obvious on the 2D data leaving no room for questions. Here this is not the case. The argument is then made on further analysis of the data which, at least in my eyes, is unclear regarding accuracy and errors. For such a small chirp demonstrated (blue lines un supplementary Fig. 9) it is of utmost importance that the analysis is carefully done and presented, as there is no doubt that the experiments will be repeated by other groups.

Do the authors integrate over the harmonic signals versus delay to produce lines like the blue line in Fig.8 (bottom)? They probably do. And then, how is the phase derived? A peak-to-peak comparison is not possible as the data are rather noisy and irregular. Do they fit a sinusoidal curve to derive the phase and if so, what is the error of the phase. Note that the claimed phase variation is less than 0.5 cycles!

I suggest the authors to revisit the analysis for all three data sets acquired at three intensities. All the data shall be shown on false color plot as well as evaluated lines and their fittings for all harmonics. The final phases shall be presented with an errorbar so that readers will be allowed to evaluate better the accuracy of the claims. If the claims made here, survive this analysis this would be ideal.

We thank the referee for reading our revised paper and considering our new experimental and theoretical results. We agree that the ability to resolve the spectral chirp, encoded in our experimental results, relies on a careful data processing and error analysis. We also agree that the original version of the resubmitted manuscript did not include a detailed description of such analysis.

In the following we provide a detailed description of the experimental results analysis, for three values of the fundamental field's intensity. Our analysis includes the following steps:

Figure 1: a, The extracted oscillation phase, using Fourier analysis, of the even (blue) and the odd (red) harmonics, as well as the oscillation of their difference ($I^{N_{odd}} - I^{N_{even}}$, black), generated using $300\mu J$ driving field. b, The 2D HHG spectrum as a function of the two colors delay, together with the phase analysis presented in (a).

1. We describe the signal processing involved in the analysis of the 2D experimental results, showing the HHG spectrum as a function of the two color delay.
2. We present the basic procedure in which we extract the two color oscillation phase, based on a Fourier analysis. We then show the direct link between the 2D measurements and the 1D spectral oscillation phase as a function of the harmonic order.
3. We provide a detailed description of the error analysis, presenting two models for extracting the error of the spectral phase.
4. We elaborate on the procedure that extracts the real part of the perturbation – revealing the spectral chirp in our experiment.

2.1 Extracting the two color phase

The main goal of figure 8 in the SI was to compare the two color perturbation in quartz to the perturbation in MgO. This comparison clearly identifies the fundamental difference in the oscillation frequency ($2\omega_0$ in quartz and $4\omega_0$ in MgO), as well as the difference in the relative phase of odd and even harmonics. Unfortunately, this spectrum contains second order diffraction of the grating, of higher harmonic order, leading to a confusing presentation of our results, where the low energy range is not properly presented. We agree with the referee that, as a consequence, linking the 2D data in figure 8 to the 1D spectral phase in figure 9 is not immediately transparent. In the revised version of the SI, we perform careful analysis, removing the second diffraction peaks and focusing on the low energy spectral range.

Figure 2: a, The extracted oscillation phase, using Fourier analysis, of the even (blue) and the odd (red) harmonics, as well as the oscillation of their difference ($I^{N_{odd}} - I^{N_{even}}$, black), generated using 350 μJ driving field. b, The 2D HHG spectrum as a function of the two colors delay, together with the phase analysis presented in (a).

Figure 3: a, The extracted oscillation phase, using Fourier analysis, of the even (blue) and the odd (red) harmonics, as well as the oscillation of their difference ($I^{N_{odd}} - I^{N_{even}}$, black), generated using 400 μJ driving field. b, The 2D HHG spectrum as a function of the two colors delay, together with the phase analysis presented in (a).

Figures 1b-3b present the 2D HHG spectrum as a function of the two color delay, applying three different values of the fundamental field's intensity. Experimentally, we resolve the oscillations of the HHG spectrum over six optical cycles (of the SH field). Here we average the signal and present its oscillations over three optical cycles (every two cycles are being averaged). We emphasize that figures 1b-3b present the raw data, averaged over two cycles, where no further analysis or data processing was performed.

In the next stage we extract, for each harmonic order, the oscillation phase. This phase is extracted via the Fourier analysis. Performing Fourier transformation over the delay scans lead to the appearance of a clear peak at $2\omega_0$. The amplitude of this peak reflects the oscillation amplitude, while its phase encodes the oscillation phase. Such Fourier analysis has been the primary tool in extracting the oscillation phase in a numerous studies performed over the past two decades [1]. A detailed description of the phase analysis is given in section 2.2.

Figure 1a-3a present the extracted phase, together with their error bars, as a function of harmonic order, for three values of the fundamental field's intensity. The red line follows the phase of the odd harmonics while the blue line follows the phase of the even harmonics (presented in fig. 9 in the SI). A careful analysis of the error is provided in section 2.3. In addition, we compare the 1D measurement together with the 2D ones by plotting the 1D spectral phases on top of the 2D measurements. The extracted phases and the maximum points of the harmonics' oscillations match perfectly. For clarity, we present both measurements in the same units – [rad]. Note that a phase variation of 0.5 rad, as the referee mentioned, corresponds to a shift of 0.08 optical cycles of the SH field.

In the next sections we will share our data analysis procedure, adding detailed error calculation for the plotted phase.

2.2 Phase extraction using Fourier analysis

We extract the oscillation phase via the Fourier analysis. Such analysis has been well established in previous studies [1–3]. Importantly, it allows us to collect the signal which oscillates with high contrast, enhancing significantly the SNR, and therefore the accuracy of the phase measurements. Our spectrometer is composed of XUV grating that reflects and splits the emitted light to different harmonics order, into the MCP. Each harmonic signal spans over several pixels, representing its spatio-spectral distribution (spatio–along the vertical axis, and spectral along the horizontal axis). In our analysis, we perform a complete spatio-spectral analysis by performing pixel-by-pixel Fourier transformation (labeled with index i). For each pixel, we isolate the Fourier component at $2\omega_0$ frequency which represents the response to the SH perturbation. We extract both the contrast of the Fourier peak ($A_{2\omega_0}(i) = |FFT(signal(\tau)_i)_{2\omega_0}|$) and the phase ($\phi_{2\omega_0}(i) = arg(FFT(signal(\tau)_i)_{2\omega_0})$), where τ is the two color delay. This step has two important advantages: We isolate the high contrast oscillating pixels while removing the contribution of other frequency components which originate from the experimental noise.

Figure 4: **Fourier transform analysis.** We plot the absolute value of the Fourier spectrum ([ar. u]) for harmonics H12,H14 and H16, showing a clear dominant peak at $2\omega_0$ frequency.

Next, we perform a coherent sum of the high contrast pixels only, for each harmonic, and extract the two color oscillation phase, presented in figure 1a-1c:

$$\Phi_{2\omega_0}(N) = \arg\left(\sum_i A_{2\omega_0}(i)e^{i\phi_{2\omega_0}(i)}\right) \quad (1)$$

We hope that our transparent and detailed data analysis answered the referees concerns.

2.3 Error calculation study

In the following we present the analysis of the experimental error. The phase, extracted from our measurement is resolved at the Fourier component of $2\omega_0$. In contrast, the experimental noise has a flat frequency response – being scattered across the entire Fourier spectral range (figure 4). Therefore, we can evaluate the mixing between the noise frequencies with our signal as:

$$X(\omega) = X_{2\omega_0}\delta(\omega - 2\omega_0) + |X_N(\omega)|e^{i\Phi_N(\omega)} \quad (2)$$

where $X_{2\omega_0} = A_{2\omega_0}e^{i\Phi_{2\omega_0}}$ being the noise-free signal at $2\omega_0$ and $|X_N(\omega)|e^{i\Phi_N(\omega)}$ is the white Gaussian noise, which is a sequence of independent identically distributed (i.i.d) variables

modeled with the distribution:

$$\begin{cases} X_N(\omega) \sim N(\mu, \sigma^2) \\ \Phi_N(\omega) \sim U(0, 2\pi) \end{cases} \quad (3)$$

Here $X_N(\omega)$ and $\Phi_N(\omega)$ are the noise amplitude and phase, respectively. In practice, according to these definitions, the phase of noise can have any value between 0 to 2π and the amplitude of the noise has a Gaussian distribution across the entire noise frequency range. To obtain the mean amplitude of the noise we average over the spectral range higher than $2\omega_0$. Then, the different moments of our $2\omega_0$ signal can be calculated as:

$$\langle \Phi(2\omega_0) \rangle = -i \int_{-\pi}^{\pi} d\Phi_N \int_{-\infty}^{\infty} dX_N f_{\Phi_N X_N}(X_N, \Phi_N) \ln \left(\frac{X(2\omega_0)}{|X(2\omega_0)|} \right) \quad (4)$$

$$\langle \Phi(2\omega_0)^2 \rangle = - \int_{-\pi}^{\pi} d\Phi_N \int_{-\infty}^{\infty} dX_N f_{\Phi_N X_N}(X_N, \Phi_N) \ln^2 \left(\frac{X(2\omega_0)}{|X(2\omega_0)|} \right) \quad (5)$$

$$\langle |X(2\omega_0)| \rangle = \int_{-\pi}^{\pi} d\Phi_N \int_{-\infty}^{\infty} dX_N f_{\Phi_N X_N}(X_N, \Phi_N) |X(2\omega_0)| \quad (6)$$

$$\langle |X(2\omega_0)|^2 \rangle = \int_{-\pi}^{\pi} d\Phi_N \int_{-\infty}^{\infty} dX_N f_{\Phi_N X_N}(X_N, \Phi_N) |X(2\omega_0)|^2 \quad (7)$$

where $\langle \Phi(2\omega_0) \rangle$ represents the average phase, $\langle \Phi(2\omega_0)^2 \rangle$ represents its second moment, $\langle |X(2\omega_0)| \rangle$ is the average amplitude, $\langle |X(2\omega_0)|^2 \rangle$ is its the second moment and $f_{\Phi_N X_N}(X_N, \Phi_N)$ being the joint probability density function:

$$f_{\Phi_N X_N}(X_N, \Phi_N) = \frac{1}{2\pi} \frac{1}{\sqrt{2\pi\sigma^2}} e^{-\frac{(X_N - \mu)^2}{2\sigma^2}} \quad (8)$$

and noticing that:

$$\Phi(\omega) = -i \ln \left(\frac{X(\omega)}{|X(\omega)|} \right) \quad (9)$$

The total error of the phase is given by:

$$\sigma_{2\omega_0}^{\Phi} = \sqrt{Var(\Phi(2\omega_0))} = \sqrt{\langle \Phi(2\omega_0)^2 \rangle - \langle \Phi(2\omega_0) \rangle^2} \quad (10)$$

and the total error of the amplitude is given by:

$$\sigma_{2\omega_0}^X = \sqrt{Var(|X(2\omega_0)|)} = \sqrt{\langle |X(2\omega_0)|^2 \rangle - \langle |X(2\omega_0)| \rangle^2} \quad (11)$$

The error analysis can be intuitively explained by representing the signal in the complex plane. Our background-free signal at $2\omega_0$ is represented by a complex number: $A_{2\omega_0} e^{i\Phi_{2\omega_0}}$ in the complex plane (Re, Im). In this plane, the white noise distribution is represented by a circle of radius μ around the origin. The experimental value of μ is calculated by averaging over the high-frequencies tail of our signal. The addition of the noise shifts the phase of our signal. The

maximal shift is achieved when the noise vector is perpendicular to the signal $\Phi_{2\omega_0} \pm \frac{\pi}{2}$ (see figure 5). In total, the addition of the noise circle leads to a shift in the signal phase which is in the range of:

$$\sigma_{2\omega_0}^{\Phi} \approx \arctan\left(\frac{\mu}{A_{2\omega_0}}\right) = \arctan\left(\frac{1}{SNR}\right) \quad (12)$$

In figure 4 we plot the Fourier spectrum of harmonic 12, 14 and 16, resolved using a driving field with an approximated intensity of using $350\mu J$ driving field. The Fourier spectrum shows clear peak at $2\omega_0$ frequency, which is almost 20 times stronger compared to the noise. The high contrast of the Fourier peak ensures the small error in its phase analysis. In figure 1a-3a, we plotted the harmonic phases including their calculated error, using the analysis presented above. Clearly the phase error is much smaller than the presented attochirp, across the harmonic range.

The revised version of the SI includes the error calculation and a summary of this analysis.

2.4 Extracting the real part of the perturbation

Finally, following the referee's comment, we present a short discussion regarding the comparison between the attochirp which was observed previously in gas phase [4] and the attochirp that was extracted in our experiment.

Two color HHG spectroscopy probes the internal dynamics by perturbing light driven electrons with a weak SH field. Such a perturbation leads to the accumulation of an additional phase: $S = S_0 + \sigma$, where S_0 represents the unperturbed action and σ represents the perturbation. Importantly, the perturbation is complex: $\sigma = \sigma^r + \sigma^i$. The imaginary component is dominated by the tunneling mechanism, while the real component is dominated by the propagation of the electron wavefunction within the bands. Since the tunneling dynamics is localized in time, σ^i is almost constant for all harmonic orders, while σ^r varies smoothly across the harmonic spectrum [1, 5]. Furthermore, the imaginary perturbation affects both even and odd harmonics simultaneously, while in the case of a real perturbation even and odd harmonics are modulated out of phase. The interplay between the real and imaginary parts of the perturbation dictates the oscillations of the harmonics with the two color delay. As the referee correctly pointed out, in the 2006 paper [4] the two color oscillations show a clear slope as a function of harmonic number. This slope can be directly observed in the raw data and does not require an additional analysis. Indeed, in Ref [4], HHG were produced in Ar, where the perturbation was dominated by the real part (odd and even harmonics oscillate out of phase). In contrast, the perturbation in our study is clearly dominated by the imaginary component – where even and odd harmonics are modulated *in phase*. Therefore, one cannot make a direct comparison between the observation in quartz and the observation in gas phase [4]. Previous studies performed by our group demonstrated the ability to separate the real and imaginary components in gas phase [1]. In our paper we perform such analysis in quartz and isolate the real part from the total complex perturbation.

In the following we describe the basic procedure to extract the complex perturbation in systems with broken inversion symmetry. The complex perturbation along two consecutive half cycles, is represented by two complex phases σ_1 and σ_2 :

$$I_{even}^{odd} \propto |\alpha_1 e^{i\varphi_1} e^{i\sigma_1(\tau)} \pm \alpha_2 e^{i\varphi_2} e^{i\sigma_2(\tau)}|^2 \quad (13)$$

where $\alpha_{1,2}$ and $\varphi_{1,2}$ are the amplitudes and phases associated with the harmonics fields emitted along the two half cycles, respectively.

In systems with broken inversion symmetry the perturbation appears in first order:

$$I^N \propto \begin{cases} 2\alpha_1\alpha_2 \cos \Delta\varphi(1 + \sigma_1^i + \sigma_2^i) + 2|\alpha_1|^2\sigma_1^i + 2|\alpha_2|^2\sigma_2^i - 2\alpha_1\alpha_2 \sin \Delta\varphi(\sigma_1^r - \sigma_2^r) + O(\sigma^2) & N \text{ is odd} \\ -2\alpha_1\alpha_2 \cos \Delta\varphi(1 + \sigma_1^i + \sigma_2^i) + 2|\alpha_1|^2\sigma_1^i + 2|\alpha_2|^2\sigma_2^i + 2\alpha_1\alpha_2 \sin \Delta\varphi(\sigma_1^r - \sigma_2^r) + O(\sigma^2) & N \text{ is even} \end{cases} \quad (14)$$

where $\Delta\phi = \phi_1 - \phi_2$.

We can isolate the real perturbation from the imaginary one by combining the pairs of neighboring harmonic orders:

$$I^{N_{odd}} - I^{N_{even}} \propto \alpha_1\alpha_2 \sin(\Delta\varphi)(\sigma_1^r - \sigma_2^r) \quad (15)$$

Therefore, the real perturbation is not directly encoded in the slope of the two color phase. It is encoded in the difference between two consecutive harmonics.

We apply this analysis and isolate the real part of the perturbation (the attochirp). Specifically, for each pair of harmonics we extract $I^{N_{odd}} - I^{N_{even}}$ and resolve its modulation phase. Figure 1a-1c presents these phases for three values of the fundamental field's intensity. We find that the modulation phase changes significantly with the harmonic order, resolving the clear fingerprint of the interband mechanism.

Figure 5: A schematic description of the estimated experimental phase error.

Comment 2:

Moreover, I see that the authors are somewhat selective on the harmonics that they analyze to make their case both on experiment as well as on theory. The even harmonics in their spectrum span from the 12th to the 21st harmonic but the authors analysis only for a few of them around the middle range. Why is this the case? In fact, the higher harmonics have even stronger signals, why would one disregard them so easily. Of course, they authors could argue, “yes but only the ones shown are relevant for the band we study”. Yet this is not true in nonlinear optics. Transitions to higher bands do also create harmonics at the energy range of the “probed bands” how do we know if these are so weak, and they do not mess up conclusions?

This is a very interesting question which goes right to the core of perturbative versus non-perturbative nonlinear optics. In perturbative nonlinear optics, nonlinear susceptibilities responsible for harmonic generation always include summation over all possible intermediate virtual states – infinitely many of them, in fact. Thus, why can one focus on a particular conduction band? The same question applies to HHG in atoms (molecules, etc) – why do we describe a particular high harmonic emission, $N\omega_0$, in strong laser fields, via one-photon recombination from a continuum state with the energy E given by the energy conservation law, $N\omega_0 = E + I_p$ (I_p being the ionization potential)? Why do we not include the integral over all other bound and continuum states which, according to perturbative nonlinear optics, should also contribute?

The rigorous answer is by now well-known: in the strong-field regime, where the electron action is very large, the corresponding integrals over all quantum paths (all states) are domi-

nated by the semi-classical paths which correspond to real excitations, selecting the state with energy $E = N\omega_0 - I_p$ out of all possible recombination states. This selection becomes possible thanks to the strong laser field and the very large action accumulated by the electron between ionization and recombination. The formal mathematical analysis of the arising Green's function integrals in the time-domain is taken using the saddle point method, leading to the concept of quantum trajectories and selecting a well-defined momentum state for the generation of a particular harmonic. This answer applies to both atoms and solids, to both inter-band and intra-band harmonics.

One should always remember the physics described by the virtual excitations and by the contributions of far-away virtual states: they describe laser-induced polarization of the active band (or state), its distortion by the instantaneous laser field. Thus, other bands are not irrelevant: they do contribute by describing field-induced modifications of the conduction band of interest. Of course, this modification is fully included in all our simulations and it does not in any way affect our concept of the inter-band Berry phase, which is accumulated during the evolution in the laser-polarized conduction and valence bands. Such band polarization might somewhat shift the emission point for a particular harmonic, due to the distortion of the band, but in no way would it affect our general derivation of the interband Berry phase.

To summarize this point: we are in the strong field regime, where we have the luxury of dealing with real excitations, which dominate virtual excitations. In contrast, virtual excitations dominate in the weak field regime, where no real excitations occur. The demarcation line between the two regimes is clearly visible in the harmonic spectra and is associated with the long harmonic plateau, which is clearly the case here.

This demarcation applies to both intra-band and inter-band harmonics: once real injection of electrons in the conduction band occurs, all harmonics are dominated by these real excitations and not by virtual ones. The most stringent test on the relative role of virtual and real excitations is done when studying the lower order harmonics, where the Brunel type lower-order nonlinearities associated with real excitations are directly competing with the Kerr-type nonlinearities of the same order, associated with virtual excitations. This transition between the two regimes has been specifically studied in solids by P Jürgens, et al, (Nature Physics, 2020) [6]. Their conclusion was summarized above.

This is why in our study we focus on the specific range of harmonics emitted from the first conduction band (harmonics H11-H17), even though our two color spectrum spans from H11 to H31: we wanted to make sure that the underlying dynamics is relatively straightforward and does not include significant real excitations to higher conduction bands. As for the virtual transitions, they are responsible for the distortion of the bands but do not change the concept of the interband Berry phase in any way. As stated above, all of this complexity is fully included in the numerical analysis.

It is worthwhile to note that propagation to higher conduction bands in the strong field regime would involve additional tunneling steps. Crucially, the concept of the interband Berry

phase introduced here will remain fully valid throughout – only the pathways would become more complex, spanning multiple bands. While these processes encode rich and complex dynamics, they are beyond the scope of this paper (see our recent publication [7] for pertinent analysis).

We have included this discussion in the SI section 3.5: "The physics of real and virtual excitations in the strong field regime and their relative importance in HHG".

Comment 3:

The comparison between theory and experiment in Fig.9 shall be done in a transparent way. For the same energy range. Clearly this not the case now and as a result the argument is not supported.

We completely agree with the referee's comment, the theoretical plots in figure 9d led to a confusion. This calculation was performed for gas phase HHG, and therefore naturally involved different experimental conditions. The main goal of this calculation was to provide an intuitive picture for the attochirp dependence on the fundamental field's intensity. We now understand that this figure led to misunderstanding, and therefore decided to remove it.

Comment 4:

The authors argue that few-cycle pulses need higher intensity and therefore these experiments will miss the interband picture. The argue that it is all interband but at high intensities the chirp reduces and apparently this makes it to look like an intraband. I do not agree with this argument. In Fig.9 C the blue line(if we believe that the error bar is the size of the point) we have an irregular curve, this is not a straight line that would be identified as interband or intraband. To me it looks more like noise.

When the HHG process is driven by few cycle pulses and shorter wavelength, the intraband mechanism becomes the dominant one (chapters 3.3 and 3.4 in the SI). Indeed, such experimental conditions successfully revealed the interband process in previous studies [8], while the intraband process played a minor role.

In our experiment, the HHG process is driven by a multicycle pulse and long wavelength. Under such conditions the interband mechanism becomes the dominant one. This is true over the entire range of fundamental field's intensities, applied in our study (figure 9 a, b and c in the SI and figures 1-3 in this letter). However, as we increase the laser intensity, the mapping between time and energy changes as well. Previous studies in gas phase HHG identified the modifications of the atto-chirp with the laser intensity. We agree with the referee that in figure 9c, the interband mechanism is not pronounced. Therefore, we performed our analysis at different field intensities, where the spectral slope of the two color phase serves as a clear fingerprint of the interband mechanism.

We fully agree that the description of the high intensity measurement was unclear. Fol-

Following the referee's comment we have modified the SI (chapter 3.2):

"Our study captures the dependence of the time-energy mapping on the fundamental field's intensity. In our measurements, the spectral slope of the modulation phase decreases as we increase the fundamental field's intensity. Such a response is in full agreement with the semi-classical interband picture. At the same time, as we increase the laser intensity, the mapping between recombination time and harmonic order becomes more flat, and thus the attochirp reduces."

Comment 5:

Last the authors argue that they used lower intensity than other studies. I would argue that they use the highest intensity in comparison to all previous time resolved studies.

We thank the referee for this comment. One important difference between our study and previous studies is related to the pulse duration. In our experimental study we apply a multicycle laser field, while previous studies apply a few cycles laser field [8, 9], allowing to generate HHG with a higher peak intensity. Another important difference with respect to ref [8] is the fundamental wavelength. As we demonstrate in the numerical analysis (chapter 3.3), the driving field wavelength plays an important role in the interplay between the interband and intraband processes. Following the referee's comment, the comparison between our study and previous studies focuses on the driving field wavelength and the pulse duration. We have included these changes in the modified version of chapter 3.4 in the SI.

Comment 6:

To my regret despite the mounting evidence that the distortions of the chirp data is dramatic when the reflective and transmission geometry are used the authors have still opted for a transmission study. Whereas their samples are some 20 μm thick, it is difficult for me to assess how important this is, given that the demonstrated degree of chirp with which the authors are supporting their argument is negligibly small. What would the authors comment on this serious issue?

We completely agree with referee – propagation effects can play an important role in solid state HHG. Previous experimental studies have identified this role as well as its influence on the underlying dynamics (SI of [8] and [10]). Acknowledging the important role of propagation effects, we performed a dedicated study to identify them via a two-color HHG spectroscopy. A detailed description of this study is provided in [11], here we summarize the main results.

HHG in solid samples are confined to be within few tens of nanometers close to the surface of the crystal. However, both the fundamental field as well as the SH field can undergo nonlinear light matter interactions as they propagate inside the crystal. Such propagation can lead to a shift in their CEP, modification of their peak intensity (see SI in [8]) or of their central

Figure 6: **Resolving pulse propagation effects via Two color spectroscopy.** a,b Comparing the two color harmonic spectrum resolved in thick (a, $200\mu m$) and thin (b, $100\mu m$) MgO crystals. This figure is adapted from [11]. c, Comparing two color high harmonic spectroscopy in transmission and reflection geometry, performed by G. Vampa et al., this figure is adapted from [10].

frequency [11]. When the interaction is driven using long pulses which are not CEP stabilized the shift in their phase plays a minor role. In addition, the accurate value of the peak intensity does not play an important role as well. Our observables are based on the accurate probing of the dynamics via the two color field. What is the influence of the nonlinear propagation on two color HHG spectroscopy?

When the interaction is induced in thick crystal ($200\mu m$), non-linear interaction leads to a frequency shift between the SH and the fundamental field such that: $\omega_{SH} = 2\omega_0 + \delta\omega$. In this case we probe the dynamics with a non-periodic field, leading to a frequency shift of the harmonic spectrum [11]. As we scan the two color delay, we modulate the central frequencies of the harmonics (figure 6a). In contrast, if the interaction is induced in thin crystal (less than $100\mu m$), the harmonics oscillate with the two color delay, while their central frequency remains unchanged (figure 6b). This observation was confirmed by another group [10], comparing transmission and reflection HHG spectroscopy (figure 6c). Moreover, a comparison between the attochirp observed in a thin crystal ($40\mu m$ thickness, [12]) in transmission geometry, and the attochirp observed in reflection geometry [10] gave identical results. Our study in quartz is performed with a thinner crystal ($20\mu m$). Scanning the two color delay modifies the harmonic signal, while their central frequency remains unchanged. This experiment reveals that in such thickness there are no signatures of propagation effects pertinent to our analysis.

3 Referee 3

After almost one year of time, the Dudovich group and collaboration partners have submitted a revised version of their manuscript on the interband Berry phase probed by attosecond interferometry. In their response letter they state that they have dedicated extensive efforts to addressing the concerns of the referees. For my part, their responses have been fully satisfactory and I now recommend publication of the beautiful manuscript in Nature. The results are extraordinary.

We thank the referee for his careful reading, valuable comments and support in the acceptance of our paper to Nature.

As a final remark, I just noticed that one of the references the authors claimed to have added is not to be found in the manuscript nor in the supplementary information.

We completely agree with the referee that this reference describes one of most important studies, performed during the early stage of the field. Therefore, we added this citation to the introduction of solid state HHG.

References

- [1] O Pedatzur, G Orenstein, V Serbinenko, H Soifer, BD Bruner, AJ Uzan, DS Brambila, AG Harvey, L Torlina, F Morales, O Smirnova, and N Dudovich. Attosecond tunnelling interferometry. *Nature Physics*, 11(10):815, 2015.
- [2] Gal Orenstein, Oren Pedatzur, Ayelet J Uzan, Barry D Bruner, Yann Mairesse, and Nirit Dudovich. Isolating strong-field dynamics in molecular systems. *Physical Review A*, 95(5):051401, 2017.
- [3] Doron Azoury, Omer Kneller, Shaked Rozen, Barry D Bruner, Alex Clergerie, Yann Mairesse, Baptiste Fabre, Bernard Pons, Nirit Dudovich, and Michael Krüger. Electronic wavefunctions probed by all-optical attosecond interferometry. *Nature Photonics*, 13(1):54–59, 2019.
- [4] N Dudovich, Olga Smirnova, J Levesque, Yu Mairesse, M Yu Ivanov, DM Villeneuve, and Paul B Corkum. Measuring and controlling the birth of attosecond xuv pulses. *Nature physics*, 2(11):781, 2006.
- [5] Dror Shafir, Hadas Soifer, Barry D Bruner, Michal Dagan, Yann Mairesse, Serguei Patchkovskii, Misha Yu Ivanov, Olga Smirnova, and Nirit Dudovich. Resolving the time when an electron exits a tunnelling barrier. *Nature*, 485(7398):343, 2012.
- [6] P Jürgens, B Liewehr, B Kruse, C Peltz, D Engel, A Husakou, T Witting, M Ivanov, MJJ Vrakking, T Fennel, et al. Origin of strong-field-induced low-order harmonic generation in amorphous quartz. *Nature Physics*, 16(10):1035–1039, 2020.
- [7] Ayelet J Uzan-Narovlansky, Álvaro Jiménez-Galán, Gal Orenstein, Rui EF Silva, Talya Arusi-Parpar, Sergei Shames, Barry D Bruner, Binghai Yan, Olga Smirnova, Misha Ivanov, et al. Observation of light-driven band structure via multiband high-harmonic spectroscopy. *Nature Photonics*, 16(6):428–432, 2022.
- [8] Manish Garg, Hee-Yong Kim, and Eleftherios Goulielmakis. Ultimate waveform reproducibility of extreme-ultraviolet pulses by high-harmonic generation in quartz. *Nature Photonics*, 12(5):291–296, 2018.
- [9] TJ Hammond, DM Villeneuve, and PB Corkum. Producing and controlling half-cycle near-infrared electric-field transients. *Optica*, 4(7):826–830, 2017.
- [10] Giulio Vampa, Jian Lu, Yong Sing You, Denitsa R Baykusheva, Mengxi Wu, Hanzhe Liu, Kenneth J Schafer, Mette B Gaarde, David A Reis, and Shambhu Ghimire. Attosecond synchronization of extreme ultraviolet high harmonics from crystals. *Journal of Physics B: Atomic, Molecular and Optical Physics*, 53(14):144003, 2020.

- [11] Gal Orenstein, Ayelet Julie Uzan, Sagie Gadasi, Talya Arusi-Parpar, Michael Krüger, Raluca Cireasa, Barry D Bruner, and Nirit Dudovich. Shaping electron-hole trajectories for solid-state high harmonic generation control. *Optics Express*, 27(26):37835–37845, 2019.
- [12] Ayelet Julie Uzan, Gal Orenstein, Álvaro Jiménez-Galán, Chris McDonald, Rui EF Silva, Barry D Bruner, Nikolai D Klimkin, Valerie Blanchet, Talya Arusi-Parpar, Michael Krüger, et al. Attosecond spectral singularities in solid-state high-harmonic generation. *Nature Photonics*, 14(3):183–187, 2020.

Reviewer Reports on the Second Revision:

Referee #2

(Remarks to the Author)

I am satisfied with the efforts of the authors and despite some remaining concerns, I now believe that this study can be published.